# Adversarially Robust Distributed Count Tracking via Partial Differential Privacy

**Zhongzheng Xiong**
School of Data Science
Fudan University
zzxiong21@m.fudan.edu.cn

**Xiaoyi Zhu**
School of Data Science
Fudan University
zhuxy22@m.fudan.edu.cn

**Zengfeng Huang**[*]
School of Data Science
Fudan University
huangzf@fudan.edu.cn

## Abstract

We study the distributed tracking model, also known as distributed functional monitoring. This model involves $k$ sites each receiving a stream of items and communicating with the central server. The server's task is to track a function of all items received thus far continuously, with minimum communication cost. For count tracking, it is known that there is a $\sqrt{k}$ gap in communication between deterministic and randomized algorithms. However, existing randomized algorithms assume an "oblivious adversary" who constructs the entire input streams before the algorithm starts. Here we consider adaptive adversaries who can choose new items based on previous answers from the algorithm. Deterministic algorithms are trivially robust to adaptive adversaries, while randomized ones may not. Therefore, we investigate whether the $\sqrt{k}$ advantage of randomized algorithms is from randomness itself or the oblivious adversary assumption. We provide an affirmative answer to this question by giving a robust algorithm with optimal communication. Existing robustification techniques do not yield optimal bounds due to the inherent challenges of the distributed nature of the problem. To address this, we extend the differential privacy framework by introducing "partial differential privacy" and proving a new generalization theorem. This theorem may have broader applications beyond robust count tracking, making it of independent interest.

## 1 Introduction

In the distributed tracking model there are $k$ sites and a single central server. Each site $i$ receives items over time in a streaming fashion and can communicate with the server. Let $S_i(t)$ be the stream that site $i$ observes up to time $t$. The sever wants to track the value of a function $f$ that is defined over the multiset union of $\{S_i(t) \mid i = 1, \cdots k\}$ at all times. The goal is to minimize the communication cost, which is defined as the total number of words communicated between the server and all sites. Due to strong motivations from distributed system applications, this model has been extensively investigated, e.g., [1, 2, 3, 4, 5, 6, 7, 8, 9, 10, 11, 12]. The theoretical study of communication complexity was initiated by [5]. Count tracking is the most basic problem in distributed tracking, where $f$ is simply the total number of items received so far. Since exact tracking requires sites to communicate every time an item arrives, incurring too much communication, the objective is to track

---

[*]Corresponding author

37th Conference on Neural Information Processing Systems (NeurIPS 2023).

an $(1 + \alpha)$-approximation. For this problem, there is a simple deterministic algorithm with $\tilde{O}\left(k/\alpha\right)^2$ communication. Huang et al. [13] proposed a randomized algorithm that achieves $\tilde{O}(\sqrt{k}/\alpha)$, but the correctness is under the assumption of an oblivious adversary, i.e., input streams are constructed in advance and are just given to sites one item at a time. In particular, the analysis assumes the input is independent of the algorithm's internal randomness. In interactive applications, this assumption is often unrealistic; the adversary can generate the next item based on previous answers from the server, making the independence assumption invalid. Moreover, the break of independence may occur unintentionally. For example, the tracking algorithm may be part of a larger system; the output of the algorithm can change the environment, from which the future input to the algorithm is generated. In such cases, we can no longer assume the independence between inputs and algorithm's internal state. The main question of this paper is: *Whether the $\sqrt{k}$ advantage of randomized count tracking algorithms is from randomness itself or the oblivious adversary assumption?*

Designing robust randomized algorithms against adaptive adversaries has received much attention recently [14, 15, 16, 17, 18, 19, 20, 21, 22, 23]. Existing research focuses on centralized settings, and in this paper, we initiate the study of adversarial robustness for distributed tracking. We provide a new randomized algorithm with communication $\tilde{O}(\sqrt{k}/\alpha)$, which provably tracks the count within an $\alpha$ relative error at all times, even in the presence of an adaptive adversary. As in [18], we utilize differential privacy (DP) to construct robust tracking algorithms. The main idea is to use DP to protect the internal randomness, so that the adversary cannot learn enough information about it to generate bad inputs. However, due to the "event-driven nature" of distributed tracking algorithms, we cannot protect the randomness in the usual sense of DP (which will be elaborated in more details below). Thus, the DP framework of [18] is not directly applicable.

To address this difficulty, a relaxed version of differential privacy, called *partial differential privacy*, is introduced. Moreover, a new generalization theorem for partial DP is proved. In partial DP we allow an arbitrary small subset of the database to be revealed, and only require the privacy of the remaining dataset is protected. The power of the new definition comes from the fact that the privacy leaked set can be chosen by the algorithm after the interaction with the adversary and the set can depend on the actual transcript. Intuitively, an interactive mechanism satisfies partial DP as long as *after* the interaction, we can always find a large subset whose privacy is protected. On the other hand, since the set we try to protect is not fixed in advance, it is subtle to give the right notion of "protecting the privacy of a large part of the data". Besides this new notion of DP, our algorithm deviates from the framework of [18] in many other details. For instance, our algorithm does not treat existing oblivious algorithms as a black box; instead, we directly modify oblivious algorithms and perform a more fine-grained privacy analysis. The contributions of this paper are summarized as follows:

1. We initiate the study of adversarially robust distributed tracking and propose the first robust counting tracking algorithm with near optimal communication.

2. To overcome the inherent challenges that arise from the distributed nature of the problem, we introduce a relaxed (and more general) version of differential privacy and prove a new generalization theorem for this notion. We believe that this new generalization theorem can be of independent interest and may have broader applications beyond count tracking.

## 1.1 Problem Definitions and Previous Results

Throughout this paper, we use $\mathcal{M}$ to denote the tracking algorithm/mechanism and $\mathcal{A}$ to denote the adversary. $N$ is used to denote the total number of items.

**The model and its event-driven nature** We assume there exists a point-to-point communication channel between each site and the server, and communication is instantaneous with no delay. It is convenient to assume that the time is divided into discrete time steps. In each step, the adversary picks one site and sends it a new item. The adversary is also allowed to skip the step and do nothing, and because of this, algorithms that can trigger new events based on the global time do not have an advantage over purely event-driven ones. For example, the server may have wanted to wait a random number of time steps before updating the output, but the adversary can always skip a large number of steps before sending the next item, which makes the waiting meaningless. That being said, it is not a restriction to only consider *event-driven* algorithms: the internal state of each site changes only when

---

[2]We use the $\tilde{O}$ notation to suppress the dependency on all polylogarithmic factors.

it receives a new item or a new message from the server, and the server's state changes only if a new message from sites arrives.

**Distributed count tracking** The goal of a count tracking algorithm $\mathcal{M}$ is to output an $(\alpha, \beta)$-approximation of the total number of items received by all sites. More specifically, with probability at least $1 - \beta$, the output of $\mathcal{M}$ is $(1 \pm \alpha)$-accurate with respect to the true answer at all time steps simultaneously. We measure the complexity of the algorithm by the total communication cost between the server and all sites. Consistent with prior research, communication cost is expressed in terms of words unless otherwise stated. We assume that any integer less than $N$ can be represented by a single word. To simplify the presentation, we assume $k \leq \frac{1}{\alpha^2}$. The case $k > \frac{1}{\alpha^2}$ can be solved with the same technique, with an extra additive $O(k \log N)$ term in the communication complexity[3].

**The adversarial model** The setting can be viewed as a two-player game between the tracking algorithm $\mathcal{M}$ and the adversary $\mathcal{A}$. At each time step $t$,

1. $\mathcal{A}$ generates a pair $u_t = (i, x)$, where $x$ is the item and $i$ is the site to send $x$ to; and $u_t$ depends on the previous items and previous outputs of $\mathcal{M}$.

2. $\mathcal{M}$ processes $u_t$ and outputs its current answer $a_t$.

The interaction between $\mathcal{A}$ and $\mathcal{M}$ generates a transcript $\pi = (u_1, a_1, u_2, a_2, \cdots)$. The objective of $\mathcal{A}$ is to cause $\mathcal{M}$ to output an incorrect answer at some step $t$.

**Existing results on count tracking** Previous results and their main ideas are discussed here.

*Deterministic complexity.* There is a simple deterministic solution to count tracking. Each site notifies the server every time their counter increases by a factor of $1 + \alpha$. Then, the server always maintains an $\alpha$-approximation to each site's counter, and their sum is an $\alpha$-approximation to the total count. It is easy to see the communication complexity of this algorithm is $O(\frac{k}{\alpha} \cdot \log N)$. We note deterministic algorithms are trivially robust to adaptive inputs. A deterministic communication lower bound of $\Omega(\frac{k}{\alpha} \cdot \log N)$ was proved in [6].

*Randomized complexity.* A randomized algorithm with $O(\frac{\sqrt{k}}{\alpha} \cdot \log N)$ communication and constant error probability was proposed in [13], which was shown to be optimal in the same paper. The main idea of their algorithm is as follows. Let $N$ be the current number of items. Unlike the above deterministic algorithm, in which each site notifies its local count according to deterministic thresholds, now the thresholds are set randomly. Let $e_i$ be the discrepancy between the true local count on site $i$ and its estimation on the server, and the total error is $e = \sum_i^k e_i$. For deterministic algorithms, all $e_i$ could have the same sign in the worst case, so on average, $e_i$ has to be less than $\alpha N / k$. On the other hand, in the randomized algorithm, each $e_i$ is a random variable. Suppose $\mathsf{Var}[e_i] \leq (\alpha N)^2 / k$ for each $i$, the total variance $\mathsf{Var}[e] \leq (\alpha N)^2$, and it is sufficient to obtain an $\alpha$-approximation with constant probability by standard concentration inequalities. Compared to deterministic estimators, now each local error $e_i$ may far exceed $\alpha N / k$.

*Robustness to adaptive inputs.* In the randomized approach described above, the analysis crucially relies on the independence assumption on $e_i$'s, since otherwise the variances do not add up and concentration inequalities cannot be applied. When the adversary is oblivious, the independence holds as long as each site uses independent random numbers. However, in the adaptive setting, this does not hold any more, and it becomes unclear whether the $\sqrt{k}$ improvement is still achievable.

## 1.2 Existing Robust Streaming Frameworks

Distributed tracking is a natural combination of streaming algorithms [24] and communication complexity [25]. Robust streaming algorithms design has become a popular topic recently and several interesting techniques have been proposed. Next, we provide a brief overview on the existing frameworks for robust streaming algorithms. Let $\mathcal{F}$ be the target function, for example, the number of distinct elements.

---

[3]Note that this extra additive $O(k \log N)$ term for $k > \frac{1}{\alpha^2}$ also exists in previous work [13] on oblivious distributed streams.

**Sketch switching** [14] Given a stream of length $N$ and an accuracy parameter $\alpha$, the flip number, denoted as $\lambda_{\alpha,N}$, is the number of times that the target function $\mathcal{F}$ changes by a factor of $1 + \alpha$. For insertion-only streams and a monotone function $\mathcal{F}$, $\lambda = \frac{1}{\alpha} \cdot \log N$. In sketch switching, we initialize $\lambda$ independent copies of an oblivious algorithm, and items in the stream are fed to all copies. The stream can be divided into $O(\lambda)$ phases; in each phase $\mathcal{F}$ increases roughly by a factor of $1 + \alpha$. During the $j$th phase, the output remains the same, and the $j$th copy is used for tracking the value. When the estimate (from the $j$th copy) has become larger than the last released output by a factor of $(1 + \alpha)$, the output is updated and $j \leftarrow j + 1$. The robustness holds because each copy is utilized no more than once, and once its randomness is revealed, the algorithm switches to a new copy. The space complexity is $\lambda$ times the space of the oblivious algorithm. Applying sketch switching on the algorithm of [13] results in a robust count tracking algorithm. However, the communication complexity increases by a factor of $\lambda$, which can be worse than the deterministic bound.

**Difference estimator** Woodruff et al. [22] refined the sketch switching approach significantly and proposed the difference estimator (DE) framework. Informally, instead of using oblivious sketches as the switching unit, the DE framework divides the stream into blocks and uses sketches on each block as switching units. Consider a part of the stream, denoted by $S$, in which the value of $\mathcal{F}$ increases from $F$ to $2F$. A technique called difference estimator (DE) was proposed to estimate the difference between values of $\mathcal{F}$ at current time $t$ and some earlier time $t_0$. The estimator is generated by maintaining $\ell = \log \frac{1}{\alpha}$ levels of DEs. In level 1, $S$ is divided into $\frac{1}{\alpha}$ blocks and the value of $\mathcal{F}$ increases by $\alpha F$ in each block. In the $j$th level, $S$ is divided into $\frac{1}{\alpha 2^{j-1}}$ blocks, and the DEs produce estimators with additive error $\alpha F$. [22] proved that for many important problems, the space complexity of such DEs is $\alpha \mathrm{Space}(\mathcal{F}) 2^{j-1}$, where $\mathrm{Space}(\mathcal{F})$ is the space complexity of $\mathcal{F}$ in the oblivious setting. Since there are $\frac{1}{\alpha 2^{j-1}}$ DEs on the $j$th level, the total space of level $j$ is $\mathrm{Space}(\mathcal{F})$ and the space is $\ell \mathrm{Space}(\mathcal{F})$ over all levels. Since blocks from all levels form a dyadic decomposition, the final estimator is the sum of $\ell$ DEs, one from each level. Thus, the total error is $\ell \alpha F$, and by adjusting $\alpha$ in the beginning by a factor of $\ell$, this produces the desired error. Applying the DE framework to distributed tracking, the communication bottleneck is from level 1, where there are $1/\alpha$ DEs. It requires synchronization at the beginning of each block, so that all sites and the server are able to agree to start a new DE. A synchronization incurs $k$ communication; thus, even ignoring other cost, the total cost is at least $\frac{k}{\alpha}$, which is no better than the deterministic bound.

**Differential privacy** Hassidim et al. [18] proposed a framework using tools from DP. Instead of switching to fresh sketches, this framework protects the randomness in the sketch using DP. The random bits in each copy of the oblivious sketch are viewed as a data point in the database, and the adversary generates an item (considered as a query in DP) at each time and observes the privatized output. By the generalization theorem of DP, if the interaction transcript satisfies DP w.r.t. the random bits, then the error in the output is close to the error of an oblivious algorithm in the non-adaptive setting (the closeness depends on the magnitude of the noise injected). Similar as in sketch switching, the output is updated only when it changes by a $(1 + \alpha)$ factor, and thus there are only $\lambda$ time steps in which the adversary observes "useful information". Therefore, it is not surprising that the sparse vector technique [26] is applied. By the advanced composition theorem of DP [27], $\tilde{O}(\sqrt{\lambda})$ independent copies of the oblivious algorithm is enough for $\lambda$ outputs. Therefore, compared with sketch switching, the space increases by only a factor of $\sqrt{\lambda}$. Attias et al. [23] gave an improvement by incorporating difference estimator to the DP framework.

However, there is a fundamental challenge in applying the DP framework to distributed tracking. As discussed in Section 1.1, all distributed tracking algorithms are essentially event-driven. Now let us focus on a time step $t$ where the server updates its output. Because of the event-driven nature, this update is triggered by the event that some site $i$ just sent a message. Similarly, site $i$ sending the message is also triggered by another event, and so on and so forth. The start of this event chain must be that the adversary sends an item to some site $j$, triggering $j$ to send the first message. This causes additional privacy leakage. For example, suppose whether to send a message is indicated by a binary function $f(n_j, r_j)$ where $r_j$ is the random number used in the tracking algorithm and $n_j$ is the local count on site $j$. At time $t$, the adversary knows $f(n_j, r_j) = 1$, which makes the algorithm have no privacy guarantee. This problem is attributed to the fact that the server can update the output only after it receives a message. So to achieve the desired level of privacy, one has to add noise to $f$ locally on each site, but the total noise from all sites can be too large.

## 1.3 Our Method

**Technical overview** In our algorithm, each site divides its stream into continuous blocks of size $\Delta = \tilde{O}(\frac{\alpha N}{\sqrt{k}})$. For each block $j$, the site draws a random integer $r_j$ with uniform distribution in $[\Delta]$. The site sends a message to the server when the number of items in a block $j$ reaches the threshold $r_j$. The server output $m\Delta$, where $m$ is the number of messages received from all sites. By a similar analysis as in [13], the estimator has $\alpha N$ additive error. To robustify this algorithm, we also use DP. To overcome the limitations of the existing DP framework, we make several critical changes. First, instead of running multiple independent copies of the oblivious algorithm, we run a single copy of the above oblivious algorithm. Secondly, we perform a more refined privacy analysis, in which each random number $r_j$ is treated as the privacy unit. Therefore, the analysis framework is quite different from [18]. Thirdly, and most importantly, we do not require the algorithm to have a privacy guarantee in the traditional sense; instead, we privatize the algorithm so that, at any time, we can always find a large set of random numbers whose privacy is protected. However, it is unclear how to change the original DP definition to capture the meaning of "protecting a large subset of the dataset", since this set depends on the current transcript, and may change at each time step. Moreover, for this weaker DP, we need to prove that the generalization theorem still holds. To this end, we introduce partial DP and prove a generalization theorem for it. We believe partial DP is quite general and will have more applications beyond robust distributed tracking. The main results of this paper is summarized in the next theorem.

**Theorem 1** (Main theorem). *With probability $1 - \delta$, our mechanism $\mathcal{M}$ comprised of Algorithm 1 and 2 outputs an $\alpha$-approximate to the total count at all times in the adversarial setting. The communication complexity is $O(\frac{C\sqrt{k}\log N}{\alpha})$ where $C = \sqrt{\left( \left(\log\sqrt{k}\right)^{1.5} + 1 \right) \cdot \log\left(\frac{8\sqrt{k}\log N}{\alpha\delta}\right)}$.*

Compared to the optimal randomized bound in the oblivious setting, the cost of handling adaptive adversaries is at most an extra factor of $C$.

## 2 Preliminaries

**Notation** Let $\mathbf{\Pi}$ be the space of all possible transcripts of the interaction between $\mathcal{M}$ and $\mathcal{A}$. We use $\Pi$ to denote the transcript random variable, $\pi \in \mathbf{\Pi}$ to denote a realization of $\Pi$. The Laplace distribution with 0 mean and $2b^2$ variance is denoted by $\mathsf{Lap}(b)$. We use the notation $S \sim \mathcal{P}^m$ to indicate that $S$ is a dataset comprised of $m$ i.i.d samples from distribution $\mathcal{P}$. The conditional distribution of $S$ given the transcript $\pi$ is represented by $\mathcal{Q}_\pi$. The query function is denoted by $q : \mathcal{X}^m \to [0, 1]$ and $q_t$ denotes the query function at time step $t$. If $q_t$ is a linear query, then $q_t(S) = \frac{1}{m}\sum_{i=1}^{d} q_{t,i}(S_i)$, where $q_{t,i} : \mathcal{X} \to [0, 1]$ is a sub-query function on a single sample. The expectation of $q$ over the distribution $\mathcal{P}^m$ is denoted by $q(\mathcal{P}^m) = \mathsf{E}_{S\sim\mathcal{P}^m}[q(S)]$. And $q(Q_\pi) = \mathsf{E}_{S\sim Q_\pi}[q(S)]$.

**Differential privacy** Let $S \in \mathcal{X}^m$ be the database that $\mathcal{M}$ needs to protect, for example in our case, the random numbers (thresholds) in $\mathcal{M}$. Denote the interaction between $\mathcal{A}$ and $\mathcal{M}$ by $I(\mathcal{M}, \mathcal{A}; S)$.

**Definition 1** (Differential Privacy). *$\mathcal{M}$ is $(\varepsilon, \delta)$-differentially private if for any $\mathcal{A}$, any two neighboring database $S \sim S' \in \mathcal{X}^m$ differing only in one position, and any event $E \subseteq \mathbf{\Pi}$, we have*

$$\Pr_{\Pi\sim I(\mathcal{M},\mathcal{A};S)}[\Pi \in E] \le e^\varepsilon \cdot \Pr_{\Pi\sim I(\mathcal{M},\mathcal{A};S')}[\Pi \in E] + \delta.$$

**Lemma 1** (Laplace Mechanism [28]). *Let $x, x' \in \mathbb{R}$ and $|x - x'| \le l$. Let $\sigma \sim \mathsf{Lap}(l/\varepsilon)$ be a Laplace random variable. For any measurable subset $E \subseteq \mathbb{R}$, $\Pr[x + \sigma \in E] \le e^\varepsilon \cdot \Pr[x' + \sigma \in E]$.*

**Private continual counting** Consider the *continual counting problem*: Given an input stream consists of $\{0, 1\}$, continual counting requires to output an approximate count of the number of 1's seen so far at every time step. Different techniques have been proposed to achieve differential privacy under continual observation [29, 30]. In this paper, we make use of the Binary Mechanism (BM) [30] (see appendix for its pseudo code).

**Theorem 2.** *([30]) BM is $(\varepsilon, 0)$-differentially private with respect to the input stream. With probability at least $1 - \delta$, the additive error is $O(\frac{1}{\varepsilon} \cdot (\log T)^{1.5} \cdot \log(\frac{T}{\delta}))$ at all time steps $t \in [T]$.*

**Remark.** *Note that although the input of BM is bits, it can directly extend to real numbers without any modification. The same privacy and utility guarantees hold.*

**Generalization by differential privacy** The generalization guarantee of differential privacy arises from adaptive data analysis. Existing research [28, 31, 32] has shown that any mechanism for answering adaptively chosen queries that is differentially private and sample-accurate is also accurate out-of-sample.

**Definition 2** (Accuracy). $\mathcal{M}$ *satisfies* $(\alpha, \beta)$*-sample accuracy for adversary* $\mathcal{A}$ *and distribution* $\mathcal{P}$ *iff*

$$\Pr_{S \sim \mathcal{P}^m, \Pi \sim I(\mathcal{M}, \mathcal{A}; S)} \left[ \max_t |a_t - q_t(S)| \geq \alpha \right] \leq \beta,$$

*where* $a_t$ *is the output of* $\mathcal{M}$ *and* $q_t$ *is the query given by* $\mathcal{A}$ *at time* $t$. $\mathcal{M}$ *satisfies* $(\alpha, \beta)$*-distributional accuracy iff*

$$\Pr_{S \sim \mathcal{P}^m, \Pi \sim I(\mathcal{M}, \mathcal{A}; S)} \left[ \max_t |a_t - q_t(\mathcal{P}^m)| \geq \alpha \right] \leq \beta.$$

Recently, Jung et al. [32] discovered a simplified analysis of the generalization theorem by introducing the posterior data distribution $\mathcal{Q}_\pi$ as the key object of interest. Through a natural resampling lemma, they showed that a sample-accurate mechanism is also accurate with respect to $\mathcal{Q}_\pi$.

**Lemma 2** ([32]). *Suppose that* $\mathcal{M}$ *is* $(\alpha, \beta)$*-sample accurate. Then for every* $c > 0$ *it also satisfies:*

$$\Pr_{S \sim \mathcal{P}^m, \Pi \sim I(\mathcal{M}, \mathcal{A}; S)} \left[ \max_t |a_t - q_t(\mathcal{Q}_\Pi)| > \alpha + c \right] \leq \frac{\beta}{c}.$$

Thus, to achieve distributional accuracy, it suffices to prove the closeness between $\mathcal{Q}_\pi$ and $\mathcal{P}^m$. Then they showed that this can be guaranteed by differential privacy.

**Lemma 3** ([32]). *If* $\mathcal{M}$ *is* $(\varepsilon, \delta)$*-differentially private, then for any data distribution* $\mathcal{P}$, *any analyst* $\mathcal{A}$, *and any constant* $c > 0$:

$$\Pr_{S \sim \mathcal{P}^m, \Pi \sim I(\mathcal{M}, \mathcal{A}; S)} \left[ \max_t |q_t(\mathcal{P}^m) - q_t(\mathcal{Q}_\Pi)| > (e^\varepsilon - 1) + 2c \right] \leq \frac{\delta}{c}.$$

## 3 Partial Differential Privacy and Its Generalization Property

In the definition of partial DP, we specify the set of data whose privacy is leaked via a mapping $f_L : \Pi \to 2^{[m]}$. Given a transcript $\pi$, partial DP guarantees privacy only on $S \setminus f_L(\pi)$. Intuitively, this means that the transcripts on two database $S, S'$ that differs only on a position $i \in S \setminus f_L(\pi)$ have similar distributions. However, $\pi$ is not known in advance, which causes trouble to this direct definition. We remedy this by first fixing $i$; then we only consider $S, S'$ that differs only on $i$ and only those events $E$ whose elements do not contain $i$ in their privacy leaked set.

**Definition 3** (Partial Differential Privacy). $\mathcal{M}$ *is* $(\varepsilon, \delta, \kappa)$*-partial differentially private, if there exists a privacy leak mapping* $f_L : \Pi \to 2^{[m]}$ *with* $\max_\pi |f_L(\pi)| \leq \kappa$, *the following holds: for any* $\mathcal{A}$, *any* $i$, *any* $S, S' \in \mathcal{X}^m$ *that differs only on the ith position, and any* $E \subseteq \Pi$ *such that* $i \notin \bigcup_{\pi \in E} f_L(\pi)$,

$$\Pr_{\Pi \sim I(\mathcal{M}, \mathcal{A}; S)} [\Pi \in E] \leq e^\varepsilon \cdot \Pr_{\Pi \sim I(\mathcal{M}, \mathcal{A}; S')} [\Pi \in E] + \delta.$$

A generalization theorem for partial DP is presented below. Generalization for linear queries suffices for our application, but this can be extended to general low-sensitivity queries.

**Theorem 3.** *For linear queries, if* $\mathcal{M}$ *satisfies* $(\varepsilon, \delta, \kappa)$ *partial differential privacy and* $\mathcal{M}$ *is* $(\alpha, \beta)$*-sample accurate, then for any data distribution* $\mathcal{P}$, *any adversary* $\mathcal{A}$, *and any constant* $c, d > 0$:

$$\Pr_{S \sim \mathcal{P}^m, \Pi \sim I(\mathcal{M}, \mathcal{A}; S)} \left[ \max_t |a_t - q_t(\mathcal{P}^m)| > \alpha + (e^\varepsilon - 1) + \frac{2\kappa}{m} + c + 2d \right] \leq \frac{\delta}{d} + \frac{\beta}{c}.$$

Compared with existing results, it has an extra term $\frac{\kappa}{m}$ in the error. This is intuitive, as the privacy leaked set contributes at most $\frac{\kappa}{m}$ error in the worst case. Following [32], it suffices to establish the low discrepancy between $\mathcal{Q}_\pi$ and $\mathcal{P}^m$. To this end, we prove the following key lemma in the appendix.

**Lemma 4.** *If $\mathcal{M}$ satisfies $(\varepsilon, \delta, \kappa)$ partial differential privacy, then for any data distribution $\mathcal{P}$, any adversary $\mathcal{A}$, and any constant $c > 0$:*

$$\Pr_{S\sim\mathcal{P}^m, \Pi\sim I(\mathcal{M},\mathcal{A};S)} \left[ \max_t |q_t(\mathcal{Q}_\pi) - q_t(\mathcal{P}^m)| > (e^\varepsilon - 1) + \frac{2\kappa}{m} + 2c \right] \le \frac{\delta}{c}.$$

The query function of interest in this paper only depends on $\hat{m}$ data points, with $\hat{m} \ll m$, at each time step. For such queries, we expect the total error to be proportional to $\hat{m}$ rather than $m$, which is formalized in the following refinement of Theorem 3, the proof of which requires only a slight modification and is included in the appendix.

**Theorem 4.** *For linear queries, if $\mathcal{M}$ satisfies $(\varepsilon, \delta, \kappa)$ partial differential privacy and $\mathcal{M}$ is $(\alpha, \beta)$-sample accurate. Further, if each linear query depends on at most $\hat{m}$ data points, then for any data distribution $\mathcal{P}$, any adversary $\mathcal{A}$, and any constant $c, d > 0$:*

$$\Pr_{S\sim\mathcal{P}^m, \Pi\sim I(\mathcal{M},\mathcal{A};S)} \left[ \max_t |a_t - q_t(\mathcal{P}^m)| > \alpha + \frac{\hat{m}}{m}(e^\varepsilon - 1 + c) + \frac{2\kappa}{m} + 2d \right] \le \frac{\delta}{d} + \frac{\beta}{c}.$$

# 4 Robust Distributed Count Tracking Algorithm

Our algorithm has multiple rounds. In each round, $N$ increases roughly by a factor of $1 + \sqrt{k}\alpha$. After a round ends, the true count is computed and sent to all sites, and then, the algorithm is reinitialized with fresh randomness. Thus, we only focus on one round, and let $N_0$ be the true count at the beginning of the round. In the algorithm, $C = \sqrt{\left( \left( \log \sqrt{k} \right)^{1.5} + 1 \right) \cdot \log \left( \frac{8\sqrt{k}}{\beta} \right)}$ and $\Delta = \frac{\alpha N_0}{8C\sqrt{k}}$.

The algorithm on site $i$ is presented in Algorithm 1. The site divides its own stream into blocks of size $\Delta$, and exactly one bit will be sent to the server in each block. The actual time of sending the bit is determined by a random threshold $r_{ij}$. Let $N_{t,i}$ be the number of items received on site $i$ from the beginning of the current round until time $t$. Let $d_{t,i} = \lfloor N_{t,i}/\Delta \rfloor$ and $e_{t,i} = N_{t,i} \mod \Delta$. Thus, $j = d_{t,i} + 1$ is the index of the current active block, and $e_{t,i}$ is the offset in this block. As per Algorithm 1, the number of bits sent by site $i$ up to time $t$ is $B_{t,i} = d_{t,i} + \mathbf{1}[r_{i,j} < e_{t,i}] \le d_{t,i} + 1$. Since $r_{i,j} \sim \mathsf{Uni}(0, \Delta)$, $\mathsf{E}[B_{t,i}] = d_{t,i} + \frac{e_{t,i}}{\Delta}$, meaning $\Delta \cdot B_{t,i}$ is an unbiased estimate of $N_{t,i}$. Let $D \in (0, \Delta)^m$ ($m = k \times k'$) be the database comprised of all sites' random numbers, i.e., $D_{ij} = r_{ij}$, considering the input generated by the adversary as queries, then the query at time $t$ can be specified as:

$$q_t(D) = \frac{1}{m} \left( \sum_{i\in[k]} d_{t,i} + \sum_{i\in[k]} \mathbf{1}[r_{i,j} < e_{t,i}] \right) = \frac{B_t}{m}, \tag{1}$$

where $B_t = \sum_{i=1}^{k} B_{t,i}$ denotes the total number of bits received by the server at time $t$. The value of $q_t$ at each time step depends on $k$ random numbers corresponding to the active blocks, which is much less than the size of $D$, which motivates the use of Theorem 4.

Let $\hat{B}_t$ be algorithm's estimate of $B_t$. Algorithm 2 consists of $\sqrt{k}$ phases; $\hat{B}_t$ remains constant in each phase and a new estimate $\hat{b}_j$ is obtained at the end of the $j$th phase via the binary mechanism. The times that the phases end are denoted by $H = \{t_1, t_2, \ldots, t_{\sqrt{k}}\}$, and for $t \in [t_j, t_{j+1})$, we have $\hat{B}_t = \hat{b}_j$. Therefore, the transcript generated by $\mathcal{M}$ and $\mathcal{A}$ is of the form $((\perp, 0), (\perp, 0), \ldots, (\top, \hat{b}_1), (\perp, \hat{b}_1), \ldots, (\top, \hat{b}_{\sqrt{k}}))$. We note, in addition to noise in BM, the only noise added for the purpose of DP is the Laplace random variable added on $T$, and an independent noise is used in each phase.

## 4.1 Privacy Analysis

In this section, we analyze the privacy of $\mathcal{M}$ for a single round, demonstrating that it satisfies $(\epsilon, 0, \sqrt{k})$-partial differential privacy with respect to the random numbers $D$ used by all sites. To achieve this, we first provide the privacy leak mapping. Note that each time the output of $\mathcal{M}$ updates, i.e, reporting $\top$, $\mathcal{A}$ knows the site $i$ it just accessed has sent a bit to the server. Then the active random

---

**Algorithm 1:** Site $i$ for a round

---

**Input:** Accuracy parameter $\alpha$, failure probability $\beta$
**Initialize:** $k' = Ck$. Generate $k'$ i.i.d random number $\{r_{i1}, r_{i2}, \cdots, r_{ik'}\}$ from the uniform
     distribution on $[\Delta]$. Initialize $c_i = 0$ to count the number of received items.

**1 When** *site $i$ receives an item*:
**2**  $c_i \leftarrow c_i + 1$
**3**  $j \leftarrow \lfloor \frac{c_i}{\Delta} \rfloor + 1$
**4**  $c_{ij} \leftarrow c_i \mod \Delta$
**5**  **if** $j > k'$ **then**
**6**    Send a signal to the server to end current round.

**7**  $c_{ij} \leftarrow c_i \mod \Delta$
**8**  **if** $c_{ij} > r_{ij}$ *for the first time* **then**
**9**    Send a bit to the server.

---

**Algorithm 2:** Server

---

**Input:** Accuracy parameter $\alpha$, failure probability $\beta$
**Initialize:** Set $\varepsilon = C/\sqrt{k}, T = 2C\sqrt{k}, \hat{T} = T + \mathsf{Lap}(\frac{4}{\varepsilon}), j = 1, a_0 = N_0$. Initialize $\sqrt{k}$
     counters $\{c_1, c_2, \cdots, c_{\sqrt{k}}\}$ to 0. Start a Binary Mechanism instance denoted as BM
     with $L = \sqrt{k}$ and $\frac{\varepsilon}{4}$ as privacy parameter.

**1 When** *receiving a bit from some site*:
**2**  $c_j \leftarrow c_j + 1$ /* $c_j$ counts the number of bits in the $j$-th phase. */
**3**  **if** $c_j < \hat{T}$ **then**
**4**    Output $(\perp, a_{j-1})$

**5**  **else**
**6**    $\hat{b}_j \leftarrow \mathsf{BM}(c_j)$ /* feeding $c_j$ as the $j$-th input to BM.    */
**7**    $a_j \leftarrow \hat{b}_j \cdot \Delta + N_0$
**8**    Output $(\top, a_j)$
**9**    **if** $j > \sqrt{k}$ **then**
**10**      Notify all sites to end the current round, collect all local counters, calculate the total
        count, broadcast it to all sites, and start the next round.
**11**    $j \leftarrow j + 1$
**12**    $\hat{T} \leftarrow T + \mathsf{Lap}(\frac{4}{\varepsilon})$

---

number $r_{ij}$ at this moment is exposed, which means there is no meaningful privacy guarantee[4]. Therefore, we need to relax the DP constraint. Given a transcript $\pi$, the privacy leaked set consists of those $r_{ij}$ that are exposed during the execution.

**Definition 4** (Privacy Leaked Set). *For a given transcript $\pi$, let $A$ be the set of time steps when output updates, i.e. $A = \{t \mid \pi_t = (\top, \cdot)\}$ where $\pi_t$ is the output at time $t$. Let $i_t$ be the site $\mathcal{A}$ chooses at time $t$. Then, $f_L(\pi) = \{(i_t, j_t) \mid t \in A(\pi), j_t = \lfloor N_{t,i_t}/\Delta \rfloor + 1\}$.*

Since there is one data point exposed in each phase, $\kappa \le \sqrt{k}$. The privacy guarantee with this privacy leak mapping is presented below.

**Lemma 5.** *For any two neighboring databases $D \sim D'$ that differ only on $(i, j)$, any transcript $\pi$ satisfying $(i, j) \notin f_L(\pi)$, our mechanism (Algorithm 1 and 2) satisfies the following inequality,*

$$e^{-\varepsilon} \cdot \Pr_{\Pi \sim I(\mathcal{M}, \mathcal{A}; D')}[\Pi = \pi] \le \Pr_{\Pi \sim I(\mathcal{M}, \mathcal{A}; D)}[\Pi = \pi] \le e^{\varepsilon} \cdot \Pr_{\Pi \sim I(\mathcal{M}, \mathcal{A}; D')}[\Pi = \pi],$$

*i.e., it satisfies $(\varepsilon, 0, \sqrt{k})$-partial differential privacy.*

---

[4]Note approximate DP is also not satisfied, since the indices $i, j$ are not random and can be manipulated by the adversary.

## 4.2 Accuracy and Communication

Observe that the size of $D$ is $m = k \times k'$, and each query $q_t$ depends only on $\hat{m} = k$ of them. In the algorithm $q_t(D)$ is estimated by $\hat{B}_t/m$. Suppose it is $(\alpha, \beta)$-sample accurate, then by Lemma 5 and Theorem 4, we have

$$\Pr_{D \sim \mathcal{P}^m, \Pi \sim I(\mathcal{M}, \mathcal{A}; D)} \left[ \max_t \left| \frac{\hat{B}_t}{m} - q_t(\mathcal{P}^m) \right| > \alpha + \frac{k}{m}(e^\varepsilon - 1 + c) + \frac{2\sqrt{k}}{m} \right] \leq \frac{\beta}{c}.$$

Note that $\varepsilon = C/\sqrt{k}$. By setting $c = 1/\sqrt{k}$, we get the following lemma.

**Lemma 6.** *For query function $q_t$ defined in equation* (1)*, if our mechanism $\mathcal{M}$ is $(\alpha, \beta)$-sample accurate for $q_t$, it is $(\frac{4C\sqrt{k}}{m} + \alpha, \sqrt{k}\beta)$-distributional accurate.*

By Theorem 2, with probability $1 - \frac{\beta}{2}$, for all $t_j \in H$, we have that:

$$\left| \hat{B}_{t_j} - B_{t_j} \right| \leq \frac{1}{\varepsilon} \cdot \left( \log \sqrt{k} \right)^{1.5} \cdot \log \left( \frac{2\sqrt{k}}{\beta} \right) \leq C\sqrt{k}, \tag{2}$$

where the second inequality is from $\varepsilon = C/\sqrt{k}$ and the definition of $C$. Denote the Laplace variables used in Algorithm 2 as $\{\sigma_1, \cdots, \sigma_{\sqrt{k}}\}$. By the union bound, with probability $1 - \frac{\beta}{2}$, $|\sigma_j| \leq \frac{\log(4\sqrt{k}/\beta)}{\varepsilon}$ for all $j \in [\sqrt{k}]$. Consider the $(j+1)$th phase. For every time step $t \in (t_j, t_{j+1})$, since $B_t - B_{t_j} \leq T + \sigma_{j+1} \leq 2C\sqrt{k} + \frac{\log(4\sqrt{k}/\beta)}{\varepsilon}$ and $\hat{B}_t = \hat{B}_{t_j}$, we have

$$\left| \hat{B}_t - B_t \right| = \left| \hat{B}_{t_j} - B_t \right| = \left| \hat{B}_{t_j} - B_{t_j} + B_{t_j} - B_t \right| \leq 3C\sqrt{k} + \frac{\log(4\sqrt{k}/\beta)}{\varepsilon} \leq 4C\sqrt{k}. \tag{3}$$

Then $\hat{B}_t/m$ is $(4C\sqrt{k}/m, \beta)$-sample accurate with respect to $q_t(D)$. By Lemma 6, it follows that $\hat{B}_t/m$ is $(8C\sqrt{k}/m, \sqrt{k}\beta)$-accurate w.r.t. $\mathsf{E}_{D \sim \mathcal{P}^m}[q_t(D)] = N_t/m\Delta$. We establish the following lemma of the accuracy guarantee.

**Lemma 7.** *With probability $1 - \sqrt{k}\beta$, for all $t$ in a round starting from $N_0$, we have $|\Delta \cdot \hat{B}_t + N_0 - (N_t + N_0)| \leq 8C\sqrt{k} \cdot \Delta = \alpha N_0$.*

For the Communication complexity, in one round, the total number of received bits by the server is $c_1 + c_2 + \ldots + c_{\sqrt{k}}$. By analysis above, with probability $1 - \beta/2$, for all $j \in [\sqrt{k}]$, we have

$$\sum_{j=1}^{\sqrt{k}} c_j \leq \sum_{j=1}^{\sqrt{k}} (\hat{T}_j + 1) \leq \sum_{j=1}^{\sqrt{k}} (T + |\sigma_j| + 1) \leq \sum_{j=1}^{\sqrt{k}} 2C \cdot \sqrt{k} = 2Ck, \tag{4}$$

$$\sum_{j=1}^{\sqrt{k}} c_j \geq \sum_{j=1}^{\sqrt{k}} \hat{T} \geq \sum_{j=1}^{\sqrt{k}} (T - |\sigma_j|) \geq \sum_{j=1}^{\sqrt{k}} C\sqrt{k} \geq Ck. \tag{5}$$

Therefore, in one round, the communication cost is upper bounded by $O(Ck)$. By (5), there are at least $Ck \cdot \Delta = \alpha N_0 \sqrt{k}/8$ items received in this round, which means $N$ increases by an $O(1 + \alpha\sqrt{k}/8)$ factor after one round. It follows that there are at most $O(\frac{\log N}{\alpha\sqrt{k}})$ rounds. Combined this with the communication cost in one round, we can conclude the final communication complexity is $O(\frac{C\sqrt{k}\log N}{\alpha})$. Now we are ready to prove Theorem 1.

*Proof of Theorem 1.* Since $N_0$ is the exact count at the beginning of a round, the Lemma 7 guarantees an $\alpha$-relative error in the round. Since the lemma holds for any round, the correctness is established. We set the failure probability as $\beta = \delta/(\sqrt{k} \cdot O(\frac{\log N}{\alpha\sqrt{k}}))$. Since there are $O(\frac{\log N}{\alpha\sqrt{k}})$ rounds, by the union bound and Lemma 7, it can be concluded that with probability $1 - \delta$, the output of $\mathcal{M}$ is an $\alpha$-approximate to $N$ at all times. The communication complexity is $O(\frac{C\sqrt{k}\log N}{\alpha})$[5]. ∎

---

[5]Note that we assume that $k \leq \frac{1}{\alpha^2}$ and thus there are at most $O(\frac{\log N}{\alpha\sqrt{k}})$ rounds. For $k > \frac{1}{\alpha^2}$, there are at most $\log N$ rounds. The communication complexity is $O(Ck \log N)$. Therefore the communication complexity for all regimes of $k$ is $O(Ck \log N + \frac{C\sqrt{k}\log N}{\alpha})$.

## 5 Conclusion

In this paper, we study the robustness of distributed count tracking to adaptive inputs. We present a new randomized algorithm that employs differential privacy to achieve robustness. Our new algorithm has near optimal communication complexity. Besides, we introduce a relaxed version of differential privacy, which allows privacy leak of some data points. Based on this definition, we prove a new generalization theorem of differential privacy, which we believe can be of independent interest and have broader applications.

## Acknowledgments and Disclosure of Funding

This work is supported by National Natural Science Foundation of China No. U2241212, No. 62276066.

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
