## A  Other Related Work

In addition to the robust streaming frameworks discussed earlier, several works in the literature have considered adversarial robustness for specific problems [14, 16, 20, 21, 33, 34, 35, 36]. [36] also studied adversarially robust sketching in a distributed setting, but only considered a non-adaptive adversary and one-shot computation. The generalization property in adaptive data analysis has been extensively studied [37, 38, 31, 32, 39, 40, 41]. Our work extends the existing studies by providing a new generalization theorem for a relaxed definition of differential privacy.

## B  Binary Mechanism

---

**Algorithm 3:** Binary Mechanism [30]

---

**Input:** A time upper bound $L$, a privacy parameter $\varepsilon$, and a stream $\sigma \in \{0,1\}^L$
**Output:** At each time step $t$, output estimate $\mathcal{B}(t)$.
**Initialize:** Each $c_i$ and $\hat{c}_i$ are (implicitly) initialized to $0$.
1  $\varepsilon' \leftarrow \varepsilon / \log L$
2  **for** $t \leftarrow 1$ **to** $L$ **do**
3      Express $t$ in binary form: $t = \sum_j \mathsf{Bin}_j(t) \cdot 2^j$.
4      Let $i := \min\{j : \mathsf{Bin}_j(t) \neq 0\}$, then $c_i \leftarrow \sum_{j<i} c_j + \sigma(t)$
5      **for** $j \leftarrow 0$ **to** $i-1$ **do**
6          $c_j \leftarrow 0, \hat{c}_j \leftarrow 0$
7      $\hat{c}_i \leftarrow c_i + \mathsf{Lap}(\frac{1}{\varepsilon'})$
8      **Output the estimate at time** $t$**:**

$$\mathcal{B}(t) \leftarrow \sum_{j:\mathsf{Bin}_j(t)=1} \hat{c}_j$$

---

## C  Missing Proofs in Section 3

Prior to presenting the missing proofs, we establish a lemma that will be utilized in subsequent proofs. To simplify notation, we will omit the $\kappa$ parameter in the definition of partial differential privacy when it is not used.

**Lemma 8.** *If $\mathcal{M}$ satisfies $(\varepsilon, \delta)$ partial differential privacy with privacy leak mapping $f_L$, given index $i \in [m]$ and data-point $x$, for any event $E$ such that $\forall \pi \in E, i \notin f_L(\pi)$, we have:*

$$\Pr_{S \sim \mathcal{P}^m, \Pi \sim I(S)}[\Pi \in E | S_i = x] \leq e^\varepsilon \Pr_{S \sim \mathcal{P}^m, \Pi \sim I(S)}[\Pi \in E] + \delta$$

*Proof.*

$$\Pr_{S \sim \mathcal{P}^m, \Pi \sim I(S)}[\Pi \in E | S_i = x] = \sum_{\boldsymbol{x} \in \mathcal{X}^m} \Pr_{S \sim \mathcal{P}^m}[S = \boldsymbol{x}] \cdot \Pr[\Pi \in E \mid S = (\boldsymbol{x}_{-i}, x)]$$

$$\leq \sum_{\boldsymbol{x} \in \mathcal{X}^m} \Pr_{S \sim \mathcal{P}^m}[S = \boldsymbol{x}] \cdot (e^\varepsilon \Pr[\Pi \in E \mid S = \boldsymbol{x}] + \delta)$$

$$= e^\varepsilon \Pr_{S \sim \mathcal{P}^m, \Pi \sim I(S)}[\Pi \in E] + \delta$$

where the inequality is from the definition of partial differential privacy. ■

### C.1  Proof of Lemma 4

*Proof of Lemma 4.* Given a transcript $\pi \in \mathbf{\Pi}$, let $t^*(\pi) = \operatorname{argmax}_t |q_t(\mathcal{Q}_\pi) - q_t(\mathcal{P}^m)|$. For an $\alpha > 0$, we define the following sets:

$$\mathbf{\Pi}_\alpha = \left\{\pi \in \mathbf{\Pi} \mid q_{t^*(\pi)}(\mathcal{Q}_\pi) - q_{t^*(\pi)}(\mathcal{P}^m) > \alpha\right\},$$

$$\mathcal{X}^+(\pi, i) = \left\{ x \in \mathcal{X} \mid \Pr_{S \sim \mathcal{Q}_\pi}[S_i = x] > \Pr_{S \sim \mathcal{P}^m}[S_i = x] \right\},$$

$$A(\pi) = \{i \in [m] \mid i \notin f_L(\pi)\},$$

$$B_\alpha^+ = \bigcup_{\pi \in \mathbf{\Pi}_\alpha} \left( \{\pi\} \times \left( \bigcup_{i \in A(\pi)} \{i\} \times \mathcal{X}^+(\pi, i) \right) \right),$$

$$\mathbf{\Pi}_\alpha^+(x, i) = \left\{ \pi \in \mathbf{\Pi} \mid (\pi, i, x) \in B_\alpha^+ \right\}.$$

Fix any $\alpha$ and suppose that $\Pr\left[|q_{t^*(\pi)}(\mathcal{Q}_\Pi) - q_{t^*(\pi)}(\mathcal{P})| > \alpha\right] > \frac{\delta}{c}$. Without loss of generality, assume that

$$\Pr\left[q_{t^*(\pi)}(\mathcal{Q}_\Pi) - q_{t^*(\pi)}(\mathcal{P}) > \alpha\right] = \Pr[\Pi \in \mathbf{\Pi}_\alpha] > \frac{\delta}{2c}. \tag{6}$$

By abuse of notation, let $I$ be the random variable obtained by uniformly sampling from $[m]$, i.e., $\Pr[I = i] = 1/m$ for all $i \in [m]$. We write $S_I$ to denote the $I$-th sample of $S \sim \mathcal{P}^m$. We consider the following comparison of two probability measures on $B_\alpha^+$:

$$\Pr_{I \otimes (S, \Pi)}[(\Pi, I, S_I) \in B_\alpha^+] - \Pr_{I \otimes S \otimes \Pi}[(\Pi, I, S_I) \in B_\alpha^+]$$

$$= \sum_{\pi \in \mathbf{\Pi}_\alpha} \Pr[\Pi = \pi] \sum_{i \in A(\pi)} \Pr[I = i] \sum_{x \in \mathcal{X}^+(\pi, i)} (\Pr[S_i = x | \Pi = \pi] - \Pr[S_i = x])$$

$$\geq \sum_{\pi \in \mathbf{\Pi}_\alpha} \Pr[\Pi = \pi] \sum_{i \in A(\pi)} \Pr[I = i] \sum_{x \in \mathcal{X}^+(\pi, i)} q_{t^*(\pi), i}(x)(\Pr[S_i = x | \Pi = \pi] - \Pr[S_i = x])$$

$$\geq \sum_{\pi \in \mathbf{\Pi}_\alpha} \Pr[\Pi = \pi] \sum_{i \in A(\pi)} \Pr[I = i] \sum_{x \in \mathcal{X}} q_{t^*(\pi)}(x)(\Pr[S_i = x | \Pi = \pi] - \Pr[S_i = x])$$

$$= \frac{1}{m} \sum_{\pi \in \mathbf{\Pi}_\alpha} \Pr[\Pi = \pi] \sum_{i \in A(\pi)} \sum_{x \in \mathcal{X}} q_{t^*(\pi), i}(x)(\Pr[S_i = x | \Pi = \pi] - \Pr[S_i = x])$$

$$\geq \frac{1}{m} \sum_{\pi \in \mathbf{\Pi}_\alpha} \Pr[\Pi = \pi] \left( m \cdot (q_{t^*(\pi)}(\mathcal{Q}_\pi) - q_{t^*(\pi)}(\mathcal{P}^m)) - |f_L(\pi)| \right) \quad (\text{By } q_{t^*(\pi), i}(x) \in [0, 1])$$

$$> \frac{1}{m} \sum_{\pi \in \mathbf{\Pi}_\alpha} \Pr[\Pi = \pi] \left( m\alpha - \max_\pi |f_L(\pi)| \right) = \Pr[\pi \in \mathbf{\Pi}_\alpha] \cdot (\alpha - \frac{\kappa}{m}).$$

On the other hand, by partial differential privacy, we have

$$\Pr_{I \otimes (S, \Pi)}[(\Pi, I, S_I) \in B_\alpha^+] - \Pr_{I \otimes S \otimes \Pi}[(\Pi, I, S_I) \in B_\alpha^+]$$

$$= \sum_{i \in [m]} \Pr[I = i] \sum_{x \in \mathcal{X}} \Pr[S_i = x](\Pr[\pi \in \mathbf{\Pi}_\alpha^+(x, i) | S_i = x] - \Pr[\Pi \in \mathbf{\Pi}_\alpha^+(x, i)])$$

$$\leq \sum_{i \in [m]} \Pr[I = i] \sum_{x \in \mathcal{X}} \Pr[S_i = x] \left( (e^\varepsilon - 1) \Pr[\Pi \in \mathbf{\Pi}_\alpha^+(x, i)] + \delta \right) \quad (\text{By Lemma 8})$$

$$= (e^\varepsilon - 1) \Pr_{I \otimes S \otimes \Pi}[(\Pi, I, S_I) \in B_\alpha^+] + \delta$$

$$\leq (e^\varepsilon - 1) \Pr[\Pi \in \mathbf{\Pi}_\alpha] + \delta$$

$$< ((e^\varepsilon - 1) + 2c) \cdot \Pr[\Pi \in \mathbf{\Pi}_\alpha], \quad (\text{By equation (6)})$$

which result in a contradiction for $\alpha \geq (e^\varepsilon - 1) + \frac{\kappa}{m} + 2c$. ∎

## C.2 Proof of Theorem 4

To prove Theorem 4, we introduce two new lemmas, with one being a variant of Lemma 4 and the other a variant of Lemma 2.

**Lemma 9.** *If $\mathcal{M}$ satisfies $(\varepsilon, \delta, \kappa)$ partial differential privacy with privacy leak function $f_L : \mathbf{\Pi} \to 2^{[m]}$. Further, if each linear query depends on at most $\hat{m}$ samples, then for any data distribution $\mathcal{P}$, any adversary $\mathcal{A}$, and any constant $c > 0$:*

$$\Pr_{S \sim \mathcal{P}^m, \Pi \sim I(\mathcal{M}, \mathcal{A}; S)} \left[ \max_t |q_t(\mathcal{Q}_\pi) - q_t(\mathcal{P})| > \frac{\hat{m}}{m} \cdot (e^\varepsilon - 1) + \frac{\kappa}{m} + 2c \right] \leq \frac{\delta}{c}. \tag{7}$$

*Proof.* The proof is slight modification of that in Lemma 4. In addition to the sets defined in the proof of Lemma 4, we introduce another set $G(\pi)$ which specifies the samples that $q_{t^*(\pi)}$ depends on, formally defined as follows.

$$G(\pi) = \{i \in [m] | i \in g(t^*(\pi), \pi)\},$$

where the function $g(t, \pi)$ is used to characterize the sample set that $q_t$ depends on given transcript $\pi$. Accordingly, the set $B_\alpha^+$ is modified to incorporate $G(\pi)$:

$$B_\alpha^+ = \bigcup_{\pi \in \mathbf{\Pi}_\alpha} \left( \{\pi\} \times \left( \bigcup_{i \in A(\pi) \cap G(\pi)} \{i\} \times \mathcal{X}^+(\pi, i) \right) \right).$$

We consider the same comparison of probability measure on $B_\alpha^+$ as that in the proof of Lemma 4. The first part is same as before,

$$\Pr_{I \otimes (S, \Pi)}[(\Pi, I, S_I) \in B_\alpha^+] - \Pr_{I \otimes S \otimes \Pi}[(\Pi, I, S_I) \in B_\alpha^+]$$

$$= \sum_{\pi \in \mathbf{\Pi}_\alpha} \Pr[\Pi = \pi] \sum_{i \in A(\pi) \cap G(\pi)} \Pr[I = i] \sum_{x \in \mathcal{X}^+(\pi, i)} (\Pr[S_i = x | \Pi = \pi] - \Pr[S_i = x])$$

$$\geq \Pr[\pi \in \mathbf{\Pi}_\alpha] \cdot (\alpha - \frac{\kappa}{m}).$$

For the second part we can get that,

$$\Pr_{I \otimes (S, \Pi)}[(\Pi, I, S_I) \in B_\alpha^+] - \Pr_{I \otimes S \otimes \Pi}[(\Pi, I, S_I) \in B_\alpha^+]$$

$$= \sum_{i \in [m]} \Pr[I = i] \sum_{x \in \mathcal{X}} \Pr[S_i = x](\Pr[\pi \in \mathbf{\Pi}_\alpha^+(x, i) | S_i = x] - \Pr[\Pi \in \mathbf{\Pi}_\alpha^+(x, i)])$$

$$\leq (e^\varepsilon - 1) \Pr_{I \otimes S \otimes \Pi}[(\Pi, I, S_I) \in B_\alpha^+] + \delta.$$

Let $B_\alpha = \bigcup_{\pi \in \mathbf{\Pi}_\alpha} \left( \{\pi\} \times \left( \bigcup_{i \in A(\pi) \cap G(\pi)} \{i\} \right) \right)$. Here comes the key observation that,

$$\Pr_{I \otimes S \otimes \Pi}[(\Pi, I, S_I) \in B_\alpha^+] \leq \Pr_{I \otimes \Pi}[(\Pi, I) \in B_\alpha] = \sum_{\pi \in \Pi_\alpha} \Pr[\Pi = \pi] \sum_{i \in A(\pi) \cap G(\pi)} \frac{1}{m} \leq \frac{\hat{m}}{m} \Pr[\Pi \in \mathbf{\Pi}_\alpha],$$

where the third inequality is from $|G(\pi)| \leq \hat{m}$. Hence,

$$\Pr_{I \otimes (S, \Pi)}[(\Pi, I, S_I) \in B_\alpha^+] - \Pr_{I \otimes S \otimes \Pi}[(\Pi, I, S_I) \in B_\alpha^+] < \left( \frac{\hat{m}}{m}(e^\varepsilon - 1) + 2c \right) \cdot \Pr[\Pi \in \mathbf{\Pi}_\alpha].$$

Combining these two parts completes the proof. ∎

Next we provide a variant of Lemma 2.

**Lemma 10.** *If $\mathcal{M}$ is $(\alpha, \beta)$-sample accurate and each linear query $q_t$ depends on at most $\hat{m}$ samples, then for any constant $c > 0$,*

$$\Pr_{S \sim \mathcal{P}^m, \Pi \sim I(\mathcal{M}, \mathcal{A}; S)} \left[ \max_t |a_t - q_t(Q_\Pi)| > \alpha + \frac{\hat{m}}{m} c \right] < \frac{\beta}{c}.$$

*Proof.* The proof presented here is a minor modification of that used in Lemma 2, provided in [32]. Let $t^*(\pi) = \arg\max_t |a_t - q_t(\mathcal{Q}_\pi)|$. The proof of Lemma 2 uses a fact that $a_{t^*(\Pi)} - q_{t^*(\Pi)}(S') - \alpha \leq 1$. Under the condition that $q_t$ only depends on $\hat{m}$ samples, it can be concluded that $a_t - \frac{1}{m} \sum_{i \in [m]} q_{t,i}(S_i) \leq \frac{\hat{m}}{m}$. Thus the fact now becomes to $a_{t^*(\Pi)} - q_{t^*(\Pi)}(S') - \alpha \leq \frac{\hat{m}}{m}$. Using this new fact in original proof of Lemma 2 can yield the inequality above. ∎

The proof of Theorem 4 is direct combination of above two lemmas.

## D   Missing Proofs in Section 4

**Remark.** *Without loss of generality, the adversary is assumed to be deterministic. This is because a randomized adversary can be regarded as a probabilistic mixture of deterministic adversaries, thereby rendering it sufficient to establish adaptive robustness against deterministic adversaries.*

### D.1 Proof of Lemma 5

*proof of Lemma 5.* As mentioned in the main text, the transcript generated by $\mathcal{M}$ and $\mathcal{A}$ is of the form $((\perp, 0), (\perp, 0), \ldots, (\top, \hat{b}_1), (\perp, \hat{b}_1), \ldots, (\top, \hat{b}_{\sqrt{k}}))$. Note that, w.l.o.g., $\mathcal{A}$ is assumed to be deterministic; thus the input generated by $\mathcal{A}$ can be fully determined by the output of $\mathcal{M}$ and thus is omitted in the transcript. Recall that Algorithm 2 consists of $\sqrt{k}$ phases and the output does not change until the end of each phase. Therefore for a given transcript $\pi$, it can be represented by $(\pi^{(1)}, \pi^{(2)}, \ldots, \pi^{\sqrt{k}})$ where $\pi^{(t)} = (\perp, \perp, \ldots, \hat{b}_t)$ is the simplified output of $t$-th phase. For notation convenience, we write $P_D[\pi]$ to denote $\Pr_{\Pi \sim I(\mathcal{M}, \mathcal{A}; D)]}[\Pi = \pi]$. Then we have:

$$P_D[\pi] = P_D[\pi^{(1)}] \cdot P_D[\pi^{(2)}|\pi^{(1)}] \cdot P_D[\pi^{(3)}|\pi^{(1)}, \pi^{(2)}] \cdots P_D[\pi^{(\sqrt{k})}|\pi^{(1)}, \pi^{(2)}, \cdots, \pi^{(\sqrt{k}-1)}] \tag{8}$$

**Privacy analysis of $P_D[\pi^{(t)}|\pi^{(1)}, \pi^{(2)}, \ldots, \pi^{(t-1)}]$.** Now we focus on one phase of Algorithm 2. Denote $(\pi^{(1)}, \pi^{(2)}, \cdots, \pi^{(t-1)})$ as $\pi^{\leq(t-1)}$. In each phase $t \in [\sqrt{k}]$, the server updates the output only when the number of received bits denoted as $c_t$ surpasses the noisy threshold $\hat{T}$. Hence the probability $P_D[\pi^{(t)}|\pi^{\leq(t-1)}]$ can be calculated as follows:

$$P_D\left[\pi^{(t)}|\pi^{\leq(t-1)}\right]$$
$$= P_D\left[c_t - 1 < \hat{T} \leq c_t\right] \cdot P_D\left[\mathsf{BM}(c_t) = \hat{b}_t | \mathsf{BM}(c_1, \ldots, c_{t-1}) = (\hat{b}_1, \ldots, \hat{b}_{t-1})\right]. \tag{9}$$

Without loss of generality, assume that $D'$ differs from $D$ at $(i, j)$ such that $D_{ij} < D'_{ij}$. If the local counter of site $i$ denoted as $n_i$ never surpasses $D_{ij}$, the output of $\mathcal{M}$ on both databases $D$ and $D'$ is identical, thus ensuring privacy. Privacy budget is only consumed when $n_i$ surpasses $D_{ij}$ or $D'_{ij}$, denoted as events $E_1$ and $E_2$, respectively. If $(i, j) \in f_L(\pi)$, then either $E_1$ or $E_2$ happens at the final time step of some phase $t$. Consequently, one of the two probability values $P_D\left[\pi^{(t)}|\pi^{\leq(t-1)}\right]$ and $P_{D'}\left[\pi^{(t)}|\pi^{\leq(t-1)}\right]$ will be zero. For instance, when $n_i$ exceeds $D_{ij}$ at the final time step of phase $t$, and as $D_{ij} < D'_{ij}$, $\mathcal{M}$ with $D'$ as input will receive no bits at this time, thus producing the same output $\perp$ as before, which results in $P_{D'}\left[\pi^{(t)}|\pi^{\leq(t-1)}\right] = 0$. To avoid this scenario, we require the condition $(i, j) \notin f_L(\pi)$. Under this condition, $P_{D'}\left[\pi^{(t)}|\pi^{\leq(t-1)}\right]$ can be computed in a similar manner to equation (9):

$$P_{D'}\left[\pi^t|\pi^{\leq(t-1)}\right]$$
$$= P_{D'}\left[c'_t - 1 < \hat{T} \leq c'_t\right] \cdot P_{D'}\left[\mathsf{BM}(c'_t) = \hat{b}_t | \mathsf{BM}(c'_1, \ldots, c'_{t-1}) = (\hat{b}_1, \ldots, \hat{b}_{t-1})\right]. \tag{10}$$

**Composition of $\sqrt{k}$ subroutines and binary mechanism.** Combining equation (8), (9) and (10) yields that

$$P_D[\pi] = (\prod_{t=1}^{\sqrt{k}} P_D[c_t - 1 < \hat{T} < c_t]) \cdot P_D\left[\mathsf{BM}(c_1, c_2, \cdots, c_{\sqrt{k}}) = (\hat{b}_1, \hat{b}_2, \cdots, \hat{b}_{\sqrt{k}})\right] \tag{11}$$

$$P_{D'}[\pi] = (\prod_{t=1}^{\sqrt{k}} P_{D'}[c'_t - 1 < \hat{T} < c'_t]) \cdot P_{D'}[\mathsf{BM}(c'_1, c'_2, \cdots, c'_{\sqrt{k}}) = (\hat{b}_1, \hat{b}_2, \cdots, \hat{b}_{\sqrt{k}})] \tag{12}$$

Since $D \sim D'$, in our mechanism, $c_t$ and $c'_t$ will differ only when $D_{ij}$ or $D'_{ij}$ is surpassed during the $t$-th phase. Since each phase uses a new counter, there exist at most two phases $t_1, t_2 \in [\sqrt{k}]$ such that $|c'_t - c_t| = 1$ and for the other phases, $c_t = c'_t$. Recall that $\hat{T} = T + \mathsf{Lap}(\varepsilon/4)$. By Lemma 1, for $t \in \{t_1, t_2\}$, we can get that

$$e^{-\frac{\varepsilon}{4}} \cdot P_{D'}\left[c'_t - 1 < \hat{T} \leq c'_t\right] \leq P_D\left[c_t - 1 < \hat{T} \leq c_t\right] \leq e^{\frac{\varepsilon}{4}} \cdot P_{D'}\left[c'_t - 1 < \hat{T} \leq c'_t\right]. \tag{13}$$

By direct calculation, we have

$$e^{-\frac{\varepsilon}{2}} \prod_{t=1}^{\sqrt{k}} P_{D'}[c'_t - 1 < \hat{T} < c'_t] \leq \prod_{t=1}^{\sqrt{k}} P_D[c_t - 1 < \hat{T} < c_t] \leq e^{\frac{\varepsilon}{2}} \prod_{t=1}^{\sqrt{k}} P_{D'}[c'_t - 1 < \hat{T} < c'_t]. \tag{14}$$

Now consider the binary mechanism. It is known from analysis above that $(c_1, \cdots, c_{\sqrt{k}})$ differs from $(c'_1, \cdots, c'_{\sqrt{k}})$ at most two positions. By Theorem 2, we have

$$e^{-\frac{\varepsilon}{2}} \cdot P_{D'}[\mathsf{BM}(c'_1, \cdots, c'_{\sqrt{k}}) = (\hat{b}_1, \cdots, \hat{b}_{\sqrt{k}})] \leq P_D[\mathsf{BM}(c_1, \cdots, c_{\sqrt{k}}) = (\hat{b}_1, \cdots, \hat{b}_{\sqrt{k}})],$$

$$P_D[\mathsf{BM}(c_1, \cdots, c_{\sqrt{k}}) = (\hat{b}_1, \cdots, \hat{b}_{\sqrt{k}})] \leq e^{\frac{\varepsilon}{2}} \cdot P_{D'}[\mathsf{BM}(c'_1, \cdots, c'_{\sqrt{k}}) = (\hat{b}_1, \cdots, \hat{b}_{\sqrt{k}})]. \quad (15)$$

Combining equation (14) and (15), we can get that

$$e^{-\varepsilon} \cdot P_{D'}[\pi] \leq P_D[\pi] \leq e^{\varepsilon} \cdot P_{D'}[\pi],$$

which completes the proof. ∎

# E  Extension to Low Sensitivity Queries

**Definition 5.** *A query $q : \mathcal{X}^m \to \mathbb{R}$ is called $\Delta$-sensitive if for all pairs of neighbouring datasets $S, S' \in \mathcal{X}^m : |q(S) - q(S')| \leq \Delta$. Note that linear queries are $(1/m)$-sensitive.*

**Lemma 11.** *If $\mathcal{M}$ satisfies $(\varepsilon, \delta, \kappa)$ partial differential privacy with privacy leak function $f_L : \Pi \to 2^{[m]}$. Further, if each $\Delta$-sensitive query $q_t$ depends on at most $\hat{m}$ samples, then for any data distribution $\mathcal{P}$, any adversary $\mathcal{A}$, and any constant $c > 0$:*

$$\Pr_{S \sim \mathcal{P}^m, \Pi \sim I(M, A; S)} \left[ \max_t |q_t(\mathcal{Q}_\pi) - q_t(\mathcal{P}^m)| > \left( e^\varepsilon - 1 + \frac{\kappa}{\hat{m}} + 4c \right) \hat{m}\Delta \right] \leq \frac{m}{\hat{m}} \cdot \frac{\delta}{c}. \quad (16)$$

*Proof.* We introduce the following useful definitions: $\bar{q}(\boldsymbol{x}_{\leq i}) = \mathbb{E}_{S' \sim \mathcal{P}^{m-i}} [q((\boldsymbol{x}_{\leq i}, S'))]$. Given a transcript $\pi \in \Pi$, let $t^*(\pi) = \arg\max_t |q_t(\mathcal{Q}_\pi) - q_t(\mathcal{P}^m)|$. We use a function $g(t, \pi)$ to specify the samples that $q_t$ depends on at time $t$ given $\pi$. For an $\alpha > 0$, we define the following sets:

$$\Pi_\alpha = \left\{ \pi \in \Pi \mid q_{t^*(\pi)}(\mathcal{Q}_\pi) - q_{t^*(\pi)}(\mathcal{P}^m) > \alpha \right\},$$
$$A(\pi) = \{ i \in [m] | i \notin f_L(\pi) \},$$
$$G(\pi) = \{ i \in [m] | i \in g(t^*(\pi), \pi) \},$$
$$\Pi_{\alpha, i} = \{ \pi \in \Pi \mid \pi \in \Pi_\alpha, i \in A(\pi) \cap G(\pi) \},$$

and for any $z \in [0, 2\Delta]$, $i \in [m]$, denote

$$\Pi_{\alpha, i, z}(\boldsymbol{x}_{\leq i}) = \left\{ \pi \in \Pi_{\alpha, i} \mid \bar{q}_{t^*(\pi)}(\boldsymbol{x}_{\leq i}) - \bar{q}_{t^*(\pi)}(\boldsymbol{x}_{\leq i-1}) > z - \Delta \right\}.$$

We will then focus on the following expectation:

$$\mathbb{E}_{S \sim \mathcal{P}^m} \left[ \sum_{\pi \in \Pi_\alpha} \Pr_{\Pi \sim I(S)}[\Pi = \pi] \sum_{i \in A(\pi) \cap G(\pi)} \left( \bar{q}_{t^*(\pi)}(S_{\leq i}) - \bar{q}_{t^*(\pi)}(S_{\leq i-1}) \right) \right].$$

On one hand, we have that

$$\mathbb{E}_{S \sim \mathcal{P}^m} \left[ \sum_{\pi \in \Pi_\alpha} \Pr_{\Pi \sim I(S)}[\Pi = \pi] \sum_{i \in A(\pi) \cap G(\pi)} \left( \bar{q}_{t^*(\pi)}(S_{\leq i}) - \bar{q}_{t^*(\pi)}(S_{\leq i-1}) \right) \right]$$

$$= \underbrace{\mathbb{E}_{S \sim \mathcal{P}^m} \left[ \sum_{\pi \in \Pi_\alpha} \Pr_{\Pi \sim I(S)}[\Pi = \pi] \sum_{i=1}^m \left( \bar{q}_{t^*(\pi)}(S_{\leq i}) - \bar{q}_{t^*(\pi)}(S_{\leq i-1}) \right) \right]}_{\text{Part I}}$$

$$- \underbrace{\mathbb{E}_{S \sim \mathcal{P}^m} \left[ \sum_{\pi \in \Pi_\alpha} \Pr_{\Pi \sim I(S)}[\Pi = \pi] \sum_{i \in f_L(\pi) \cap G(\pi)} \left( \bar{q}_{t^*(\pi)}(S_{\leq i}) - \bar{q}_{t^*(\pi)}(S_{\leq i-1}) \right) \right]}_{\text{Part II}}$$

$$- \underbrace{\mathbb{E}_{S \sim \mathcal{P}^m} \left[ \sum_{\pi \in \Pi_\alpha} \Pr_{\Pi \sim I(S)}[\Pi = \pi] \sum_{i \in \bar{G}(\pi)} \left( \bar{q}_{t^*(\pi)}(S_{\leq i}) - \bar{q}_{t^*(\pi)}(S_{\leq i-1}) \right) \right]}_{\text{Part III}}.$$

Note that query $q_{t^*(\pi)}$ does not depend on the $i$th data for those $i \in \bar{G}(\pi)$. Therefore $\forall i \in \bar{G}(\pi), \bar{q}_{t^*(\pi)}(S_{\leq i}) - \bar{q}_{t^*(\pi)}(S_{\leq i-1}) = 0$, which means the third part equals to zero. We then bound the first two parts separately.

$$\text{Part I} = \mathop{\mathsf{E}}_{S \sim \mathcal{P}^m} \left[ \sum_{\pi \in \mathbf{\Pi}_\alpha} \mathop{\Pr}_{\Pi \sim I(S)} [\Pi = \pi] \sum_{i=1}^m \left( \bar{q}_{t^*(\pi)}(S_{\leq i}) - \bar{q}_{t^*(\pi)}(S_{\leq i-1}) \right) \right]$$

$$= \mathop{\mathsf{E}}_{S \sim \mathcal{P}^m} \left[ \sum_{\pi \in \mathbf{\Pi}_\alpha} \mathop{\Pr}_{\Pi \sim I(S)} [\Pi = \pi] \left( \bar{q}_{t^*(\pi)}(S) - \bar{q}_{t^*(\pi)}(\mathcal{P}^m) \right) \right]$$

$$= \sum_{\pi \in \mathbf{\Pi}_\alpha} \Pr[\Pi = \pi] \left( \bar{q}_{t^*(\pi)}(\mathcal{Q}_\pi) - \bar{q}_{t^*(\pi)}(\mathcal{P}^m) \right) > \alpha \cdot \Pr[\Pi \in \mathbf{\Pi}_\alpha],$$

$$\text{Part II} \leq \mathop{\mathsf{E}}_{S \sim \mathcal{P}^m} \left[ \sum_{\pi \in \mathbf{\Pi}_\alpha} \mathop{\Pr}_{\Pi \sim I(S)} [\Pi = \pi] \cdot |f_L(\pi) \cap G(\pi)| \Delta \right] \leq \mathop{\mathsf{E}}_{S \sim \mathcal{P}^m} \left[ \sum_{\pi \in \mathbf{\Pi}_\alpha} \mathop{\Pr}_{\Pi \sim I(S)} [\Pi = \pi] \cdot |f_L(\pi)| \Delta \right]$$

$$\leq \kappa \Delta \mathop{\mathsf{E}}_{S \sim \mathcal{P}^m} \left[ \mathop{\Pr}_{\Pi \sim I(S)} [\Pi \in \mathbf{\Pi}_\alpha] \right] = \kappa \Delta \cdot \Pr[\Pi \in \mathbf{\Pi}_\alpha].$$

Combining the results together, we have

$$\mathop{\mathsf{E}}_{S \sim \mathcal{P}^m} \left[ \sum_{\pi \in \mathbf{\Pi}_\alpha} \mathop{\Pr}_{\Pi \sim I(S)} [\Pi = \pi] \sum_{i \in A(\pi) \cap G(\pi)} \left( \bar{q}_{t^*(\pi)}(S_{\leq i}) - \bar{q}_{t^*(\pi)}(S_{\leq i-1}) \right) \right] > \Pr[\Pi \in \mathbf{\Pi}_\alpha] \cdot (\alpha - \kappa \Delta).$$

On the other hand, we consider that

$$\mathop{\mathsf{E}}_{S \sim \mathcal{P}^m} \left[ \sum_{\pi \in \mathbf{\Pi}_\alpha} \mathop{\Pr}_{\Pi \sim I(S)} [\Pi = \pi] \sum_{i \in A(\pi) \cap G(\pi)} \left( \bar{q}_{t^*(\pi)}(S_{\leq i}) - \bar{q}_{t^*(\pi)}(S_{\leq i-1}) \right) \right]$$

$$= \sum_{i=1}^m \mathop{\mathsf{E}}_{S \sim \mathcal{P}^m} \left[ \sum_{\pi \in \mathbf{\Pi}_{\alpha,i}} \mathop{\Pr}_{\Pi \sim I(S)} [\Pi = \pi] \left( \bar{q}_{t^*(\pi)}(S_{\leq i}) - \bar{q}_{t^*(\pi)}(S_{\leq i-1}) \right) \right].$$

Now for each coordinate $i$, we have

$$\mathop{\mathsf{E}}_{S \sim \mathcal{P}^m} \left[ \sum_{\pi \in \mathbf{\Pi}_{\alpha,i}} \mathop{\Pr}_{\Pi \sim I(S)} [\Pi = \pi] \left( \bar{q}_{t^*(\pi)}(S_{\leq i}) - \bar{q}_{t^*(\pi)}(S_{\leq i-1}) + \Delta \right) \right]$$

$$= \mathop{\mathsf{E}}_{S \sim \mathcal{P}^m} \left[ \int_0^{2\Delta} \mathop{\Pr}_{\Pi \sim I(S)} [\Pi \in \mathbf{\Pi}_{\alpha,i,z}(S_{\leq i})] dz \right]$$

$$\leq \mathop{\mathsf{E}}_{S \sim \mathcal{P}^m, Y \sim \mathcal{P}} \left[ \int_0^{2\Delta} \left( e^\epsilon \mathop{\Pr}_{\Pi \sim I(S^{i \leftarrow Y})} [\Pi \in \mathbf{\Pi}_{\alpha,i,z}(S_{\leq i})] + \delta \right) dz \right]$$

$$= \mathop{\mathsf{E}}_{S \sim \mathcal{P}^m, Y \sim \mathcal{P}} \left[ e^\epsilon \sum_{\pi \in \mathbf{\Pi}_{\alpha,i}} \mathop{\Pr}_{\Pi \sim I(S^{i \leftarrow Y})} [\Pi = \pi] \left( \bar{q}_{t^*(\pi)}(S_{\leq i}) - \bar{q}_{t^*(\pi)}(S_{\leq i-1}) + \Delta \right) + 2\Delta \delta \right]$$

$$= \mathop{\mathsf{E}}_{S \sim \mathcal{P}^m, Y \sim \mathcal{P}} \left[ e^\epsilon \sum_{\pi \in \mathbf{\Pi}_{\alpha,i}} \mathop{\Pr}_{\Pi \sim I(S)} [\Pi = \pi] \left( \bar{q}_{t^*(\pi)}(S_{\leq i}^{i \leftarrow Y}) - \bar{q}_{t^*(\pi)}(S_{\leq i-1}) + \Delta \right) + 2\Delta \delta \right].$$

The inequality holds due to partial differential privacy and the fact that position $i$ does not belong to the privacy leak set. $S^{i \leftarrow Y}$ stands for $(S_1, \ldots, S_{i-1}, Y, S_{i+1}, \ldots, S_m)$. Therefore, in the last equality we have that $(S, Y)$ and $(S^{i \leftarrow Y}, S_i)$ are distributed identically. Since $Y \sim \mathcal{P}$ and is independent of $\Pi$, we have that

$$\mathop{\mathsf{E}}_{Y \sim \mathcal{P}} \left[ \bar{q}_{t^*(\pi)}(S_{\leq i}^{i \leftarrow Y}) \right] = \bar{q}_{t^*(\pi)}(S_{\leq i-1}),$$

$$\mathop{\mathsf{E}}_{S \sim \mathcal{P}^m} \left[ \sum_{\pi \in \mathbf{\Pi}_{\alpha,i}} \mathop{\Pr}_{\Pi \sim I(S)} [\Pi = \pi] \left( \bar{q}_{t^*(\pi)}(S_{\leq i}) - \bar{q}_{t^*(\pi)}(S_{\leq i-1}) + \Delta \right) \right]$$

$$\leq \mathop{\mathsf{E}}_{S \sim \mathcal{P}^m} \left[ e^\epsilon \mathop{\Pr}_{\Pi \sim I(S)}[\Pi \in \mathbf{\Pi}_{\alpha,i}]\Delta + 2\Delta\delta \right] = (e^\epsilon \Pr[\Pi \in \mathbf{\Pi}_{\alpha,i}] + 2\delta)\,\Delta.$$

Subtracting $\Pr[\Pi \in \mathbf{\Pi}_{\alpha,i}]\Delta$ on both sides gives

$$\mathop{\mathsf{E}}_{S \sim \mathcal{P}^m} \left[ \sum_{\pi \in \mathbf{\Pi}_{\alpha,i}} \mathop{\Pr}_{\Pi \sim I(S)}[\Pi = \pi] \left( \bar{q}_{t^*(\pi)}(S_{\leq i}) - \bar{q}_{t^*(\pi)}(S_{\leq i-1}) \right) \right] \leq ((e^\epsilon - 1)\Pr[\Pi \in \mathbf{\Pi}_{\alpha,i}] + 2\delta)\,\Delta.$$

Here comes the key observation that

$$\sum_{i=1}^m \Pr[\Pi \in \mathbf{\Pi}_{\alpha,i}] = \sum_{i=1}^m \sum_{\pi \in \mathbf{\Pi}_\alpha, i \in A(\pi) \cap G(\pi)} \Pr[\Pi = \pi] = \sum_{\pi \in \mathbf{\Pi}_\alpha} \sum_{i \in A(\pi) \cap G(\pi)} \Pr[\Pi = \pi]$$

$$\leq \sum_{\pi \in \mathbf{\Pi}_\alpha} |A(\pi) \cap G(\pi)| \Pr[\Pi = \pi] \leq \hat{m} \sum_{\pi \in \mathbf{\Pi}_\alpha} \Pr[\Pi = \pi] \leq \hat{m} \Pr[\Pi \in \mathbf{\Pi}_\alpha].$$

Now using these results, we can achieve that

$$\mathop{\mathsf{E}}_{S \sim \mathcal{P}^m} \left[ \sum_{\pi \in \mathbf{\Pi}_\alpha} \mathop{\Pr}_{\Pi \sim I(S)}[\Pi = \pi] \sum_{i \in A(\pi)} \left( \bar{q}_{t^*(\pi)}(S_{\leq i}) - \bar{q}_{t^*(\pi)}(S_{\leq i-1}) \right) \right]$$

$$\leq \sum_{i=1}^m ((e^\epsilon - 1)\Pr[\Pi \in \mathbf{\Pi}_{\alpha,i}] + 2\delta)\,\Delta \leq \left( (e^\epsilon - 1)\Pr[\Pi \in \mathbf{\Pi}_\alpha] + 2\delta\frac{m}{\hat{m}} \right) \hat{m}\Delta.$$

In summary, we obtain both an upper and a lower bound of the expectation. Suppose that $\Pr\left[ |q_{t^*(\pi)}(\mathcal{Q}_\Pi) - q_{t^*(\pi)}(\mathcal{P})| > \alpha \right] > \frac{m}{\hat{m}} \cdot \frac{\delta}{c}$. Without loss of generality, assume that

$$\Pr\left[ q_{t^*(\pi)}(\mathcal{Q}_\Pi) - q_{t^*(\pi)}(\mathcal{P}) > \alpha \right] = \Pr[\Pi \in \mathbf{\Pi}_\alpha] > \frac{1}{2}\frac{m}{\hat{m}} \cdot \frac{\delta}{c}.$$

By this assumption, we reach

$$\Pr[\Pi \in \mathbf{\Pi}_\alpha]\cdot(\alpha - \kappa\Delta) < \left( (e^\epsilon - 1)\Pr[\Pi \in \mathbf{\Pi}_\alpha] + 2\delta\frac{m}{\hat{m}} \right) k'\Delta < \Pr[\Pi \in \mathbf{\Pi}_\alpha]\left( (e^\epsilon - 1) + 4c \right)\hat{m}\Delta.$$

This results in a contradiction for $\alpha \geq \left( e^\varepsilon - 1 + \frac{\kappa}{\hat{m}} + 4c \right)\hat{m}\Delta$. ∎

Combining this Lemma with Lemma 2 yields the generalization theorem for low sensitivity queries.

**Theorem 5.** *If $\mathcal{M}$ satisfies $(\varepsilon, \delta, \kappa)$ partial differential privacy and each $\Delta$-sensitive query $q_t$ depends on at most $\hat{m}$ data. Then for any data distribution $\mathcal{P}$, any adversary $\mathcal{A}$, and any constant $c, d > 0$:*

$$\mathop{\Pr}_{S \sim \mathcal{P}^m, \Pi \sim I(M,A;S)} \left[ \max_t |a_t - q_t(\mathcal{P}^m)| > \left( e^\varepsilon - 1 + \frac{\kappa}{\hat{m}} + 4c \right)\hat{m}\Delta + \alpha + d \right] \leq \frac{m}{\hat{m}} \cdot \frac{\delta}{c} + \frac{\beta}{d}.$$

# F    Simplified Proof for $\varepsilon$-Partial Differential Privacy

Similar as that in [32], we also provide a simplified proof of a generalization theorem for $(\varepsilon, 0)$-partial differential privacy. The results are summarized in Lemma 12 and Theorem 6.

**Lemma 12.** *If $\mathcal{M}$ satisfies $(\varepsilon, 0, \kappa)$ partial differential privacy with privacy leak function $f_L : \mathbf{\Pi} \to 2^{[m]}$, then for any data distribution $\mathcal{P}$, any transcript $\pi \in \mathbf{\Pi}$, any linear query $q$, and any $\eta > 0$:*

$$\mathop{\Pr}_{S \sim \mathcal{Q}_\pi} \left[ |q(S) - q(\mathcal{P}^m)| \geq (e^\epsilon - 1) + \frac{\kappa}{m} + \sqrt{\frac{2\ln(2/\eta)}{m}} \right] \leq \eta.$$

*Proof.* Recall that for linear queries $q(S) = \frac{1}{m}\sum_{i=1}^m q_i(S_i)$ and $q(\mathcal{P}^m) = \mathsf{E}_{S \sim \mathcal{P}^m}[q(S)] = \frac{1}{m}\sum_{i=1}^m \mathsf{E}_{S_i \sim \mathcal{P}}[q_i(S_i)] = \frac{1}{m}\sum_{i=1}^m q_i(\mathcal{P})$. By the same proof in [32], we can construct a martingale and show concentration by Azuma's inequality. More specifically, define random variables

$X_i = q_i(S_i) - \mathsf{E}[q_i(S_i)|S_{<i}]$ and let $Z_i = \frac{1}{m}\sum_{j=1}^{i} X_j$. Then the sequence $Z_0 = 0, Z_1, \cdots, Z_m$ forms a martingale and $|Z_i - Z_{i-1}| \le \frac{1}{m}$. By Azuma's inequality, it can be concluded that:

$$\Pr_{S \sim Q_\pi}\left[\left|\frac{1}{m}\sum_{i=1}^{m} q_i(S_i) - \frac{1}{m}\sum_{i=1}^{m}\mathsf{E}\left[q_i(S_i)\mid S_{<i}\right]\right| \ge t\right] \le 2\exp\left(\frac{-t^2 m}{2}\right). \tag{17}$$

If $\mathcal{M}$ satisfies differential privacy strictly, [32] shows that $\mathsf{E}[q_i(S_i)|S_{<i}]$ is close to $q_i(\mathcal{P})$ for all $i \in [m]$. However, there exists privacy leak in our mechanism. For a given transcript $\pi \in \mathbf{\Pi}$, we partition the underlying samples into two sets and examine each case separately.

1. $i \notin f_L(\pi)$. Fix any realization $\mathbf{x}$ and consider $\mathsf{E}[q_i(S_i)|S_{<i} = \mathbf{x}_{<i}]$, we have

$$\mathop{\mathsf{E}}_{S \sim Q_\pi}[q_i(S_i)|S_{<i} = \mathbf{x}_{<i}] = \sum_x q_i(x) \cdot \Pr_{S \sim \mathcal{P}^m}[S_i = x|\Pi = \pi, S_{<i} = \mathbf{x}_{<i}]$$

$$= \sum_x q_i(x) \cdot \frac{\Pr_{S \sim \mathcal{P}^m}[\Pi = \pi|S_i = x, S_{<i} = \mathbf{x}_{<i}] \cdot \Pr_{S \sim \mathcal{P}^m}[S_i = x]}{\Pr_{S \sim \mathcal{P}^m}[\Pi = \pi|S_{<i} = \mathbf{x}_{<i}]}.$$

By the definition of partial differential privacy, we have that

$$e^{-\varepsilon} \le \frac{\Pr_{S \sim \mathcal{P}^m}[\Pi = \pi|S_i = x, S_{<i} = \mathbf{x}_{<i}]}{\Pr_{S \sim \mathcal{P}^m}[\Pi = \pi|S_{<i} = \mathbf{x}_{<i}]} \le e^{\varepsilon}.$$

Hence, we can conclude that for $i \notin f_L(\pi)$,

$$e^{-\varepsilon} q_i(\mathcal{P}) \le \mathop{\mathsf{E}}_{S \sim Q_\pi}[q_i(S_i)|S_{<i}] \le e^{\varepsilon} q_i(\mathcal{P}).$$

2. $i \in f_L(\pi)$. Since there are at most $\kappa$ samples that may leak privacy, combined with the fact that $q_i \in [0, 1]$, the error on these samples can be bounded as follows.

$$-\kappa \le \sum_{i \in f_L(\pi)}\left(\mathsf{E}[q_i(S_i)|S_{<i}] - q_i(\mathcal{P})\right) \le \kappa.$$

By analysis above, we can conclude that

$$\sum_{i \in [m]}\left(\mathsf{E}[q_i(S_i)|S_{<i}] - q_i(\mathcal{P})\right) = \sum_{i \notin f_L(\pi)}\left(\mathsf{E}[q_i(S_i)|S_{<i}] - q_i(\mathcal{P})\right) + \sum_{i \in f_L(\pi)}\left(\mathsf{E}[q_i(S_i)|S_{<i}] - q_i(\mathcal{P})\right),$$

$$(1 - e^{\varepsilon}) \cdot m - \kappa \le \sum_{i \in [m]}\left(\mathsf{E}[q_i(S_i)|S_{<i}] - q_i(\mathcal{P})\right) \le (e^{\varepsilon} - 1) \cdot m + \kappa.$$

By equation (17), we can get that with probability $1 - \eta$,

$$-\sqrt{2m\ln(2/\eta)} \le \sum_{i=1}^{m} q_i(S_i) - \sum_{i=1}^{m}\mathsf{E}\left[q_i(S_i)\mid S_{<i}\right] \le \sqrt{2m\ln(2/\eta)}.$$

Combining this with analysis above yields that

$$\frac{1}{m}\left|\sum_{i=1}^{m}\left(q_i(S_i) - q_i(\mathcal{P})\right)\right| \le \sqrt{\frac{2\ln(2/\eta)}{m}} + (e^{\varepsilon} - 1) + \frac{\kappa}{m},$$

which completes the proof. ∎

A generalization theorem follows directly from Lemma 12. The proof is same as that of Theorem 23 in [32].

**Theorem 6.** *If $\mathcal{M}$ satisfies $(\varepsilon, 0, \kappa)$ partial differential privacy and $\mathcal{M}$ is $(\alpha, \beta)$-sample accurate. Then for any data distribution $\mathcal{P}$, any adversary $\mathcal{A}$, any linear query $q_t$, and any constant $c, d > 0$:*

$$\Pr_{S \sim \mathcal{P}^m, \Pi \sim I(\mathcal{M}, \mathcal{A}; S)}\left[\max_t |a_t - q_t(\mathcal{P}^m)| > \alpha + \sqrt{\frac{2\ln(2/\eta)}{m}} + (e^{\varepsilon} - 1) + \frac{\kappa}{m}\right] \le \beta + \eta.$$

If a linear query $q$ only depends on $\hat{m}$ samples, then $q(S) - q(\mathcal{P}^m) = \frac{\hat{m}}{m} \cdot \left( q'(S) - q'(\mathcal{P}^{\hat{m}}) \right)$ where $q' = \frac{1}{\hat{m}} \sum_{j \in \{i_1, i_2, \cdots, i_{\hat{m}}\}} q_j(S_j)$. Applying Lemma 12 with $q'$ yields that,

$$\Pr_{S \sim \mathcal{Q}_\pi} \left[ |q(S) - q(\mathcal{P}^m)| \geq \frac{\hat{m}}{m} \cdot (e^\epsilon - 1) + \frac{\kappa}{m} + \frac{1}{m} \sqrt{2\hat{m} \ln(2/\eta)} \right] \leq \eta.$$

Thus in this case, we can get the following theorem.

**Theorem 7.** *If $\mathcal{M}$ satisfies $(\varepsilon, 0, \kappa)$ partial differential privacy and $\mathcal{M}$ is $(\alpha, \beta)$-sample accurate. Further, if each linear query $q_t$ depends on at most $\hat{m}$ samples. Then for any data distribution $\mathcal{P}$, any adversary $\mathcal{A}$, and any constant $c, d > 0$:*

$$\Pr_{S \sim \mathcal{P}^m, \Pi \sim I(\mathcal{M}, \mathcal{A}; S)} \left[ \max_t |a_t - q_t(\mathcal{P}^m)| > \alpha + \frac{\hat{m}}{m} \cdot (e^\epsilon - 1) + \frac{\kappa}{m} + \frac{1}{m} \sqrt{2\hat{m} \ln(2/\eta)} \right] \leq \beta + \eta.$$