# OpenReview forum: "Adversarially Robust Distributed Count Tracking via Partial Differential Privacy"
_NeurIPS.cc/2023/Conference — NeurIPS 2023 poster_

### Official Review · Reviewer_MMAV · 2023-06-15

**Soundness:** 3 good
**Presentation:** 3 good
**Contribution:** 4 excellent
**Rating:** 8
**Confidence:** 4

**Summary:**

This paper studies adversarial robustness for the problem of counting on distributed streams, where the input may be adaptive based on previous outputs by a protocol across $k$ sites and the objective is for a central server to output an additive $\alpha m$ approximation to the stream of length $m$. When the input is oblivious, there is a randomized protocol that uses $\tilde{O}(\sqrt{k}/\alpha)$ communication, which improves upon a deterministic protocol using $\tilde{O}(k/\alpha)$ communication. However, it is not clear how the protocol behaves when the input is adaptive. This paper gives a protocol that uses $\tilde{O}(\sqrt{k}/\alpha)$ communication, showing there is essentially no overhead for adversarial inputs.

The algorithm works by having each site split its input into blocks and notify the central server if its number of updates within a server block surpasses a certain randomized threshold. Whereas a previous work in adversarially robust streaming used sparse vector technique and differential privacy to protect the internal randomness of the algorithm, the challenge is that distributed streams are event based, and so sparse vector technique will not work. Instead, this paper proposes a new notion of partial differential privacy and shows that not only does it handle generalization with respect to the sample accuracy, but also the distributional accuracy. Thus the output is close to its expectation, which circumvents a complicated analysis of the estimator.

**Strengths:**

There are multiple solid contributions by this paper.
- Firstly, it introduces the problem of adversarial robustness to distributed streams.
- Secondly, it gives an algorithm that shows there is no overhead for the counting problem, which is not clear at all a priori.
- Thirdly, the paper introduces the concept of partial differential privacy and proves interesting generalization properties of partial differential privacy. The paper notes that partial differential privacy may be of independent interest.
- There is a good summary of related work for adversarial robustness, the central techniques in those works, and why they do not apply in this setting.
- Adversarial robustness on streams is a relevant topic for both the machine learning and theory communities.

**Weaknesses:**

- The communication cost is actually $O(k+\sqrt{k}(\log N)/\alpha)$ rather than $O(\sqrt{k}(\log N)/\alpha)$, so the regime of improvement is for $k>\frac{1}{\alpha^2}$, though it should be noted that this same weakness is also present in previous work [13] on oblivious distributed streams. Nevertheless this should be corrected in the formal guarantees.

- While there is a good summary of related work for adversarial robustness and why they do not apply in this setting, it is not clear why natural modifications to the techniques of [13] would not work.

Minor comments:

- It should be noted that $\tilde{O}$ is being used to suppress polylogarithmic terms in $N$, rather than $\tilde{O}(f)=O(f)\cdot\text{polylog(f)}$, due to the notation $\tilde{\sqrt{k}/\alpha}$ to mean $O(\sqrt{k}(\log N)/\alpha)$.

- [18] is "Hassidim" rather than "Hasidim" (note the correct spelling in [21])

- Line 294: "the the"

**Questions:**

Is it possible to think of partial differential privacy to protect a certain set of elements in the manner of carefully distributing the privacy budget as in [20]?

EDIT: I acknowledge receipt of the author response and am currently choosing to maintain my score.

---

> ### Author Rebuttal · Authors · 2023-08-09
>
>
> 1. The communication cost is actually $O(k+ \sqrt{k}(\log⁡ N)/\alpha)$ rather than $O(\sqrt{k}(\log⁡ N)/\alpha)$,  so the regime of improvement is for $k > 1/\alpha^2$, though it should be noted that this same weakness is also present in previous work [13] on oblivious distributed streams. Nevertheless this should be corrected in the formal guarantees.
>
>    It is true that the communication cost is $O(k\log N + \sqrt{k}\log N/\alpha )$; thanks for pointing this out and we will make this clear in the revision. However, $O(k\log N + \sqrt{k}\log N/\alpha )$ is still always better than $O({k}\log N/\alpha)$, which is the deterministic bound.
>
> 2. While there is a good summary of related work for adversarial robustness and why they do not apply in this setting, it is not clear why natural modifications to the techniques of [13] would not work.
>
>    Thanks for the question. It is not impossible to obtain robust algorithms by applying natural modifications to oblivious algorithms. However, to prove such modifications actually achieve robustness can be very challenging, since we need to rule out all adaptive adversarial strategies and the posterior distribution of the random bits after adaptive interactions can be extremely difficult to analyze. This is a common challenge in both distributed and centralized streaming settings and is the main motivation behind techniques such as the DP framework, which provide analytical tools for proving robustness.
>
> 3. Minor comments.
>
>    Thanks for your comments. We will correct them in the revision.
>
> 4. Is it possible to think of partial differential privacy to protect a certain set of elements in the manner of carefully distributing the privacy budget as in [20]?
>
>    There are similarities between the two techniques. However, they are quite different in several aspects. First of all, the overall framework in [20] still require full DP to apply the "DP to generalization" theorem. On the contrary, our partial DP notion is a strict relaxation on that. The relaxation is essential, since it is difficult, if not impossible, to achieve full DP without sacrificing communication in our setting, due to the event-driven nature.
>
>    More technically, [20] applies the revised sparse vector technique of [KMS]: it removes data points participating in too many "meaningful queries" to protect their privacy. However, in our problem the privacy budget of each data point is not "smooth": a single query can make it completely revealed; and we cannot anticipate such events beforehand and remove the data before it is revealed. This motivates us to define partial DP which allows privacy leakage on a small subset. In some sense, the privacy leaked set plays the same role as the deleted set in their framework, except that the deleted set in their framework is still private while the privacy leaked set is not.
>
>    For applications of DP where only their generalization is concerned, partial DP can be an alternative tool to the revised sparse vector technique of [KMS]. It is easier to use since it doesn't need to do privacy budget control explicitly and there is no need to track which data points are compromised. For example, we may simply apply the standard sparse vector technique without removing data points and prove that it satisfies partial DP. This can potentially simplified the design and analysis of robust algorithms.
>
>
>  [HKM+] Avinatan Hassidim, Haim Kaplan, Yishay Mansour, Yossi Matias, Uri Stemmer. Adversarially robust streaming algorithms via differential privacy.
>
> [KMS] Haim Kaplan, Yishay Mansour, Uri Stemmer, The Sparse Vector Technique, Revisited.

---

### Official Review · Reviewer_W7US · 2023-07-07

**Soundness:** 3 good
**Presentation:** 4 excellent
**Contribution:** 4 excellent
**Rating:** 7
**Confidence:** 2

**Summary:**

The paper provides an algorithm or the distributed count tracking that both enjoys the communication advantage from randomization and is robust to adaptive adversaries. Another contribution of the paper is that it introduces a new *partial differential privacy* definition and completes related generalization theorem that can have potential broader implications.

**Strengths:**

**Originality** The proposed method is novel by creatively designing a new privacy criteria and randomizing the communication to account for adaptive adversaries.
The new definition *partial DP* and the generalization theorem will be useful for other stream data setting.

**Quality**  The theoretical analysis looks sound; the authors can verify the efficient tradeoffs among privacy, accuracy, and communication of the proposed method.

**Clarity** The problem is well-defined and clearly stated. The method and the analysis are connected smoothly.

**Significance** The problem being solved is important. The new DP definition can have broader impact for other private ML tasks.

**Weaknesses:**

**No experiments** Could you please justify why you think experiments are not needed in your paper? Because you claim your proposed method is the first robust tracking algorithm in the domain, it seems important to offer some baseline experiments to help future work.

**Questions:**

1. I am curious what are potential scenarios that your newly proposed partial DP may be important? It would be helpful to suggest some.

2. Do you think it would be a great idea to briefly mention how the algorithm you propose can be easily adapted for other robust distributed tracking beyond the count tracking?

**Limitations:**

The authors do not discuss the limitations.

---

> ### Author Rebuttal · Authors · 2023-08-09
>
> 1. No experiments.
>
>    Our primary focus is on the theoretical aspects of the problem. Since the robustness of an algorithm against all possible adversaries, which is a desirable property, can only be established by theoretical proofs, we think a thorough theoretical study is of higher priority. However, we acknowledge the value of incorporating experimental studies as a complementary perspective to our theoretical findings. We will make efforts to conduct experimental studies and make the code available to facilitate future work.
>
> 2. What are the potential scenarios that your newly proposed partial DP may be important?
>
>    * Firstly, partial DP should be an important tool for designing robust distributed tracking algorithms, as the event-driven nature poses challenges in achieving differential privacy with a moderate level of noise. Even in the centralized streaming setting, where DP is achievable, partial DP could potentially simplify the design and analysis of robust algorithms, since it is always easier to satisfy partial DP than DP.
>
>    * From a broader perspective, we believe that the generalization property of partial DP can find wider applications beyond robust algorithms design. In any application using the generalization property of differential privacy, one can consider replacing DP with the less stringent partial DP.
>
> 3. Do you think it would be a great idea to briefly mention how the algorithm you propose can be easily adapted for other robust tracking beyond the count tracking ?
>
>    Thank you for the suggestion. Count tracking is a key building block for tracking heavy hitters and quantiles [1], so we think by combining techniques from [1], it is not difficult to adapt our algorithm for such problems. On the other hand, many other tracking problems can differ significantly , it remains a good research problem whether the algorithm in this work can be directly applied to other tracking problems. However, the challenge of privacy protection posed by the event-driven nature is inherent in all tracking problems, and thus we believe that the generalization theorem of partial differential privacy developed in this work can be a helpful tool in general. We will provide a briefly discussion on this in the revision.
>
> [1] Zengfeng Huang, Ke Yi, and Qin Zhang. Randomized algorithms for tracking distributed count,
> frequencies, and ranks.

---

### Official Review · Reviewer_8gDd · 2023-07-09

**Soundness:** 3 good
**Presentation:** 2 fair
**Contribution:** 2 fair
**Rating:** 6
**Confidence:** 3

**Summary:**

This work studies adversarially robust count tracking in a distributed computation setting. In particular the work considers a setting with an adaptive adversary that can choose future inputs based on the outputs produced by the server so far. This is in contrast to prior work on distributed count tracking that considers only oblivious adversaries that must choose the entire input stream before the session begins.

This is the first work to consider robustness against active adversaries in the distributed setting. However, prior work in the central model [18] proves that a differentially private algorithm can be used to extend obliviously robust algorithms to ones that are robust against active adversaries while retaining similar accuracy guarantees. Taking inspiration from [18], this work uses a notion related to differential privacy, and a distributed algorithm that is robust against oblivious adversaries, to construct a distributed algorithm that is robust against active adversaries. Instead of a general black box reduction like in [18] (one that can be applied to any robust algorithm), this work directly modifies some oblivious algorithms to obtain algorithms that are robust against active adversaries. The work additionally argues that the communication complexity of their algorithms is near optimal.

**Strengths:**

Like the authors mention in their manuscript, and support with a clear discussion: it is very important to study the robustness of streaming algorithms in the presence of  active adversaries since this is a more realistic model of the settings in which these algorithms operate.

This work provides a distributed count tracking algorithm that is robust against active adversaries while also achieving near optimal communication complexity and accuracy.

**Weaknesses:**

This work states that the event driven nature of distributed tracking algorithms prevents them from protecting randomness in the DP sense. However, they do not provide a concrete claim or any formal proof  of this fact. The only explanation of this claim is a brief paragraph on page 4 of the manuscript, making it difficult to understand what exactly the authors mean by the statement and also making it difficult to verify if their claim is true. Given that this claim is the central motivation for defining partial differential privacy, I would have hoped for a more thorough treatment of the claim. They also do not define differential privacy against adaptive attackers in the distributed setting.

This paper doesn’t provide a discussion of the privacy consequences of using their partial DP notion to provide privacy in distributed settings. Since this work defines a new notion that they call privacy, I believe it is essential to provide at least some discussion of:
- the limitations of this new notion

- whether the definition should be expected to provide real world privacy for computations

Minor comments:
- Despite switching between discussions of additive and multiplicative error throughout the manuscript, this work does not define accuracy anywhere and often uses the phrase accuracy without an ‘additive’ or ‘multiplicative’ quantifier.

- This work often states that they use ‘differential privacy’ to achieve robustness against active adversaries in the distributed setting, therefore conflating differential privacy with the weaker notion defined in this work.

- The way that the authors phrase the statement that their algorithm does not treat existing oblivious algorithms as a black-box is somewhat misleading. The statement seems to imply that a non black-box use of prior algorithms is generally desirable; However, while a non black-box use of algorithms is sometimes necessary to achieve optimal guarantees – a black-box use of prior algorithms often leads to a more generalizable result.

**Questions:**

Regarding the claim: “the event driven nature of distributed tracking algorithms prevents them from protecting randomness in the DP sense.”

 - [With respect to the discussion on pg 4] Why does the fact that the server can update the output only after it receives an input imply that you cannot provide privacy against active attackers in the distributed setting? Why should we expect this to be different than in the central setting?

- In the above claim, are you considering adaptive differential privacy as in  https://arxiv.org/abs/2112.00828?

- Could you explain,  formally, what exactly your claim is?

**Limitations:**

This paper doesn’t provide a discussion of the privacy consequences of using their partial DP notion to provide privacy in distributed settings. Since this work defines a new notion that they call privacy, I believe it is essential to provide at least some discussion of:
- the limitations of this new notion

- whether the definition should be expected to provide real world privacy for computations

Finally, if this notion does not indeed provide intuitive individual or group privacy in the setting the authors consider, this should be made very clear – perhaps by calling the notion something other than ’privacy’.

---

> ### Author Rebuttal · Authors · 2023-08-09
>
> 1. This paper doesn’t provide a discussion of the privacy consequences of using their partial DP notion to provide privacy in distributed settings. Since this work defines a new notion that they call privacy, I believe it is essential to provide at least some discussion of: the limitations of this new notion? whether the definition should be expected to provide real world privacy for computations?
>
>    * We emphasize that the robust tracking problem studied is not a privacy protection problem per se: we only want to track the answer accurately against an adaptive adversary, and there is *no* privacy concern. According to the framework of [1], what we really need is to prove the algorithm satisfies a generalization result; and informally speaking, DP implies generalization [1], which is the reason why DP is used. A key contribution of this paper is that we show DP is actually not necessary for this purpose, and a relaxed version, namely partial DP, is sufficient.
>
>
>    * So, whether partial DP is a good notion to provide privacy and what are its limitations in privacy sensitive applications are not so relevant to the theme of the paper, since we only care about its generalization performance, for which we provide a rigorous proof. One can call the new notion another name without using the word privacy, but we feel partial DP is appropriate because of its intimate connection to DP.
>
>
>
> 2. About the event driven claim: Why does the fact that the server can update the output only after it receives an input imply that you cannot provide privacy against active attackers in the distributed setting? Why should we expect this to be different than in the central setting?
>
>    * We only claim that, due to the event driven nature, running a centralized DP algorithm on the server does not necessarily provide DP guarantee for all sites; the paragraph on page 4 provide a counterexample. So the framework in [1] is not immediately applicable. However, we do not make a claim that "this imply that you cannot provide privacy against active attackers in the distributed setting". In fact, as we have discussed on page 4, sites can protect privacy by adding noise locally, except that this results in a bigger error.
>    * Basically, we provide intuitive reasons why achieving full DP and the same error as in the oblivious setting simultaneously is difficult. In line 171 of page 4, the statement is "there is a fundamental challenge...", where we do not claim that this is theoretically impossible; it could be achievable but seems extremely challenging.
>    * On the other hand, we show that partial DP with optimal error is achievable, and more importantly, partial DP has the same robustness guarantee as DP, and thus the fundamental challenge mentioned above is circumvented. To sum up, it is difficult, if not impossible, to apply the old method [1] in our setting (but we have never made this a mathematical statement); The contribution of this paper is that we provide a new method which circumvents this difficulty and solves the problem optimally.
>
> 3. Are you considering adaptive differential privacy as in https://arxiv.org/abs/2112.00828?
>
>    No. The difference is twofold. First, in their setting, the sensitive dataset is received in a streaming fashion, while in our problem, the actual dataset we want to protect is the random bits used in the randomized tracking algorithm (not the data streams), which are all given in the beginning. Second, they only consider the centralized setting, which is quite different from our distributed setting.
>
> 4. Multiplicative or additive error?
>
>    In Section 1.1 (line 86) of our paper, we introduce the objective of our problem, which is to output an estimate within $(1\pm \alpha)$ multiplicative error. Our analysis consider each round separately. In each round the count N increase by at most a factor of 2 (see line 273), and thus an $(1\pm \alpha)$ multiplicative error is equivalent to an $O(\alpha) N$ additive error in this round. So we focus on additive error in each round, which is easier to analyze. We appreciate the feedback and will make this clearer in the revised version.
>
> 5. Conflating differential privacy with the weaker notion.
>
>    Thanks for pointing it out. We will try to rephrase these statements to avoid such misunderstandings.
>
> 6. Statements about non black-box use.
>
>    Sorry for the misunderstandings. We point out the non black-box use is only to emphasize that the techniques and analyses used in our work are very different from existing ones with black-box use.  We agree that a block-box use of priors algorithms is more generalizable. We will try to rephrase these statements in order to mitigate any potential misunderstandings.
>
>
> [1] Hassidim A, Kaplan H, Mansour Y, et al. Adversarially robust streaming algorithms via differential privacy[J]. Advances in Neural Information Processing Systems, 2020, 33: 147-158.

---

> > ### Comment · Reviewer_8gDd · 2023-08-18
> >
> > I thank the authors for their responses! I have a better understanding of the work and have updated my review accordingly.

---

### Official Review · Reviewer_5Raw · 2023-07-09

**Soundness:** 4 excellent
**Presentation:** 3 good
**Contribution:** 3 good
**Rating:** 5
**Confidence:** 3

**Summary:**

The paper considers a distributed count tracking problem introduced at PODS 2012: k parties receive (possibly adversarial) updates to a common counter, and need to communicate with a server to maintain an approximation of the total number of updates received. The goal is to minimize communication from the parties to the server. In PODS 2012 it was shown that communication proportional to $\sqrt{k}$ is sufficient, improving on a simple deterministic algorithm for which communication is proportional to $k$. The $\sqrt{k}$ bound was also shown optimal up to polylogarithmic factors. However, the upper bound crucially assumes an oblivious adversary, such that the inputs to the parties may not depend on the estimates reported so far. The present paper shows that this assumption can be removed. The idea is to use techniques from differential privacy, but existing techniques do not apply, so the paper proposes a new version of these techniques that is applicable in the distributed setting.

UPDATE: After reading the review of MMAV and the corresponding rebuttal I am increasing my score slightly.

**Strengths:**

- Resolves a decade-old open problem regarding one of the most natural problems in distributed functional monitoring
- Introduces interesting new techniques for dealing with adaptive, adversarial inputs, which should arguably be the standard model for any dynamic algorithm claiming to be robust


**Weaknesses:**

- The paper does not really try to argue for relevance to machine learning. Unless this is addressed, it would probably be of interest to only a small fraction of NeurIPS attendees.
- Though adversarial robustness and the connection to differential privacy has been studied in NeurIPS before, this has been in the context of streaming algorithms, which are probably closer to machine learning applications. The paper feels more like a database theory paper than a NeurIPS paper.

**Questions:**

- Can you elaborate on why the results of the paper are of interest for a machine learning audience? Are there, for example, potential implications for federated learning?


**Limitations:**

I would have liked the model to be described more clearly. If my understanding is correct, only communication from sites to the server is counted, while the server is able to broadcast estimates "for free" to the sites. It is unclear in what settings this kind of asymmetry in accounting for communication is a good model.

---

> ### Author Rebuttal · Authors · 2023-08-09
>
> 1. The paper does not really try to argue for relevance to machine learning.  Can you elaborate on why the results of the paper are of interest for a machine learning audience?
>     * First, as we have repeatedly emphasized in our paper, a key contribution of this paper is introducing a new notion of differential privacy and proving a new generalization theorem, which are both highly relevant topics in machine learning. In particular, as we all know, the study of generalization is a central topic in machine learning research.
>      * Second, as another reviewer said, adversarial robustness on streams is a relevant topic for the machine learning community. Recently, a lot of research papers, e.g., [1, 2, 3], on this topic have been published in machine learning conferences like NeurIPS and ICML. We extend centralized streaming to the distributed streaming setting, which is no less relevant, since many real world machine learning scenarios are inherently distributed and the data are received as multiple streams. Moreover, the counting problem studied in this paper is even more basic and ubiquitous than many other problems studied in prior work.
>
>    [1] Hassidim A, Kaplan H, Mansour Y, et al. Adversarially robust streaming algorithms via differential privacy[J]. Advances in Neural Information Processing Systems, 2020, 33: 147-158.
>
>    [2] Cherapanamjeri Y, Nelson J. On adaptive distance estimation[J]. Advances in Neural Information Processing Systems, 2020, 33: 11178-11190.
>
>    [3] Cohen E, Lyu X, Nelson J, et al. On the robustness of countsketch to adaptive inputs[C]//International Conference on Machine Learning. PMLR, 2022: 4112-4140.
>
> 2. Only communication from sites to the server is counted. while the server is able to broadcast estimates 'for free' to the sites.
>
>    * We do count the communication from the server. The communication cost is defined in Section 1.1 (Problem Definitions and Previous Results), which is the **total communication cost between the the server and all sites** (line 88). In fact, we only assume there is a point-to-point communication channel between each site and the server (line 73), so each broadcast is counted as $k$ point-to-point messages, which incur a cost of $O(k)$.
>
>    * Our algorithm has $O(\log N/\alpha \sqrt{k})$ rounds (see section 4.2 Accuracy and Communication , line 334), and there is only one broadcast in each round (see Algorithm 2, line 10), and thus the total communication cost for broadcasting is also $O(\sqrt{k}\log N)/\alpha$, which is bounded by the total communication cost we claimed. We will add explicit explanations in the future version to make this clearer. Thanks for pointing it out.

---

> > ### Comment · Reviewer_5Raw · 2023-08-10
> > **Follow-up questions to rebuttal**
> >
> > Thanks for clarifying the model, which has made it easier to understand the results. Part of my confusion has to do with parameter restrictions. When you say that the protocol has $O(\log N/\alpha\sqrt{k})$ rounds you implicitly assume that $k = O(\log^2(N) \alpha^2)$. Since $N$ is increasing, if $k$ is fixed this condition does not seem to hold initially, so it would seem that you need an additive term to cover the part of the protocol for which $N$ is not large enough to make the condition hold?
> >
> > I acknowledge that related problems have appeared in ML venues, but I still feel that the problem studied here is a step further away from ML applications than these works.

---

> > > ### Author Response · Authors · 2023-08-15
> > > **The condition on k**
> > >
> > > Thanks for the reply. We didn't make it very clear in the submission. Here are some details on the number of rounds.
> > >
> > > In each round the number of items increases by roughly a factor of $(1+\alpha \sqrt{k})$. Thus, to count the number of rounds we consider two cases: 1. $\alpha \sqrt{k}<0.5$ and 2. $\alpha \sqrt{k}\ge 0.5$. For case 1, we have $k< 0.25/\alpha^2$ and the number of rounds is $O(\log N/\alpha \sqrt{k})$. For case 2, the number of rounds is $O(\log N)$. In both cases, the number of rounds is well-defined and thus there is no restrictions on the value of $k$. The communication cost per round is $O(Ck)$, and therefore the total cost is always bounded by $O((Ck+C\sqrt{k}/\alpha)\log N)$. We will clarify this in the revision.

---

### Decision · Program_Chairs · 2023-09-21

**Decision:**

Accept (poster)

**Comment:**

This paper studies adversarial robustness for the problem of counting on $k$ distributed streams, where the input may be adaptive based on previous outputs by a protocol across the streams sites and the objective is for a central server to output an additive approximation
Prior work handles the case when the adversary is oblivious. This paper gives a protocol for the adaptive adversary that uses $\tilde O(\sqrt{k})$ communication communication, matching the rate known for the oblivious case. This resolves a problem open since 2012. The analysis relies on techniques from differential privacy that have been introduced in the context of adaptive data analysis. In particular, it proposes a new notion of partial differential privacy and shows that not only does it handle generalization with respect to the sample accuracy, but also the distributional accuracy.